# Bi-directional Weakly Supervised Knowledge Distillation for Whole Slide Image Classification

**Linhao Qu**[*]
Digital Medical Research Center,
School of Basic Medical Science,
Fudan University
lhqu20@fudan.edu.cn

**Xiaoyuan Luo**[*]
Digital Medical Research Center,
School of Basic Medical Science,
Fudan University
19111010030@fudan.edu.cn

**Manning Wang**[†]
Digital Medical Research Center,
School of Basic Medical Science,
Fudan University
mnwang@fudan.edu.cn

**Zhijian Song**[†]
Digital Medical Research Center,
School of Basic Medical Science,
Fudan University
zjsong@fudan.edu.cn

## Abstract

Computer-aided pathology diagnosis based on the classification of Whole Slide Image (WSI) plays an important role in clinical practice, and it is often formulated as a weakly-supervised Multiple Instance Learning (MIL) problem. Existing methods solve this problem from either a bag classification or an instance classification perspective. In this paper, we propose an end-to-end weakly supervised knowledge distillation framework (WENO) for WSI classification, which integrates a bag classifier and an instance classifier in a knowledge distillation framework to mutually improve the performance of both classifiers. Specifically, an attention-based bag classifier is used as the teacher network, which is trained with weak bag labels, and an instance classifier is used as the student network, which is trained using the normalized attention scores obtained from the teacher network as soft pseudo labels for the instances in positive bags. An instance feature extractor is shared between the teacher and the student to further enhance the knowledge exchange between them. In addition, we propose a hard positive instance mining strategy based on the output of the student network to force the teacher network to keep mining hard positive instances. WENO is a plug-and-play framework that can be easily applied to any existing attention-based bag classification methods. Extensive experiments on five datasets demonstrate the efficiency of WENO. Code is available at https://github.com/miccaiif/WENO.

## 1 Introduction

Histopathological images play an important role in cancer diagnosis and prognosis prediction [23, 33, 32, 20, 27, 26], and they can be scanned by digital slide scanners into Whole Slide Images (WSIs), which facilitates the development of deep learning-based automatic analysis techniques. However, there are two challenges in deep learning-based WSI analysis. First, WSIs have huge resolutions, typically reaching $100,000 \times 100,000$ pixels, and thus cannot be directly input into deep models. For this reason, WSIs are usually tiled into many small patches for processing. Second, fine-grained

---

[*]Equal Contribution.

[†]Corresponding Authors. All authors are also from Shanghai Key Lab of Medical Image Computing and Computer Assisted Intervention.

36th Conference on Neural Information Processing Systems (NeurIPS 2022).

(patch-level) annotation is very time-consuming and labor-intensive, and usually pathologists can only provide slide-level labels, so traditional supervised learning methods cannot be directly used. Therefore, WSI classification is often formulated as a deep multiple instance learning (MIL) problem, which is a weakly supervised learning paradigm [32, 28, 4, 37].

In the MIL paradigm, each WSI is considered as a bag, and the patches cut out of it are considered as its instances. If a bag is negative, all the instances in it are negative, while if a bag is positive, at least one positive instance exists in it. Typically, deep MIL-based WSI classification performs two main tasks: bag classification and instance classification, which are used for automatic clinical diagnosis and positive region localization, respectively.

Currently, deep MIL-based WSI classification methods can be mainly classified into instance-based approach and bag-based approach. The instance-based approach [2, 5] first trains an instance classifier and then aggregates the predictions of each instance in a bag to obtain the bag prediction [7, 34, 35]. Because of the lack of patch-level labels, it is not known which patches are truly positive, so instance-based approach needs to select some patches from positive slides and assign them pseudo positive labels for training an instance classifier. The main problem of this approach is that the **pseudo instance labels contain a lot of noise**, which limits the performance of the trained instance classifier, and thus leads to inaccurate instance and bag classification.

The more common bag-based approach [13, 9, 40, 38, 31, 18, 29, 19, 30, 22] first extracts the features of each instance in a bag and aggregates the instance features to obtain the bag feature using a trainable attention mechanism. Then, a bag classifier is trained in a supervised manner. During inference, the bag classifier is used to perform bag classification and the attention scores can be utilized to measure the contribution of each instance to bag classification so as to perform the instance classification. However, this bag-based approach suffers from two serious problems: **1) Poor performance of instance classification.** Different positive instances have different levels of difficulty of being recognized. The classifier is trained on bag-level loss, so it can correctly recognize a positive bag by giving high attention weights to only a few easily recognized positive instances, while the harder ones are ignored. This makes the network tend to accurately classify only a few easy positive instances in the positive bags and lack the motivation to accurately classify the hard instances [31]. **2) Bias of the bag classifier.** For the same reason, the trained bag-classifier will have difficulty to generalize on bags that contain only hard positive instances.

To address the above issues, we propose an end-to-end **we**akly supervised k**no**wledge distillation framework (**WENO**) for whole slide image classification. WENO integrates a bag classifier and an instance classifier in a knowledge distillation framework to mutually improve the performance of both classifiers by effective knowledge transfer between them. To the best of our knowledge, the concept of weakly supervised knowledge distillation is proposed for the first time. Figure 1 concisely illustrates the existing knowledge distillation paradigms and the principle of WENO proposed in this paper. Specifically, WENO contains a bag classifier based on the attention mechanism as the teacher branch and an instance classifier as the student branch. **To address the problem of poor instance classification performance**, we use the normalized attention scores of the teacher network as the soft pseudo labels of the instances in the positive bags to train the student branch through knowledge distillation, which alleviates the noisy pseudo label problem in previous instance-based methods. At the same time, since all the instances in negative bags are negative, they are also used in training the student branch to further improve its instance classification performance. Notably, different from common knowledge distillation methods in Figure 1 (a) and (b), we not only train the student branch but also train the teacher branch using the bag labels of WSIs. Moreover, we share the instance feature extractor of the student and the teacher to further enhance the knowledge transfer between the two branches. **To further address the problem of bias in the bag classifier**, we propose a hard positive instance mining (HPM) strategy. In particular, we use the knowledge learned by the student instance classifier to construct bags with less easy instances, forcing the teacher network to keep mining hard positive instances in positive bags.

**The advantages of our method are:**

• We propose a weakly supervised knowledge distillation framework, WENO, for WSI classification. WENO integrates a bag classifier and an instance classifier in a knowledge distillation framework to mutually improve the performance of both classifiers by two-way knowledge transfer between them.

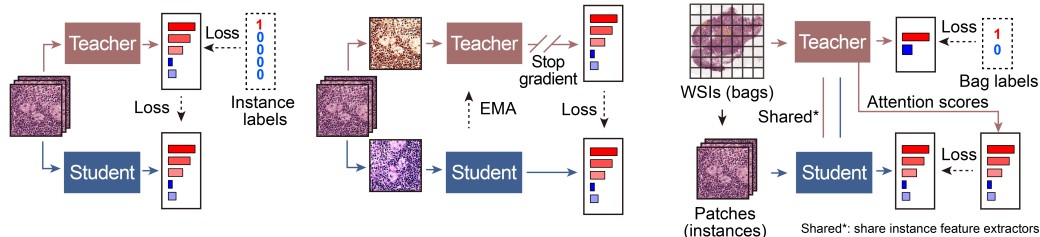

Figure 1: Architecture of common existing knowledge distillation frameworks and the proposed weakly supervised knowledge distillation framework. In traditional supervised knowledge distillation (a), the teacher network is trained in advance and it keeps unchanged during training the student. Knowledge is distilled from the teacher to the student. In recent self-knowledge distillation (b), such as DINO [3], the teacher has the same architecture with the student and it is not trainable but updated from the student. Two-way knowledge transfer exists between the teacher and the student. In comparison, in our proposed framework (c), the teacher is a bag-classifier and it is also trained with weak slide-level labels. Knowledge is distilled from teacher to student by providing pseudo instance labels using attention scores of the teacher and knowledge transfer from the student to the teacher is achieved by sharing instance feature extractors between them.

- We further propose a hard positive instance mining (HPM) strategy to force the teacher network to continuously learn and mine hard positive instances in positive bags, thus alleviating the problem of focusing on easy positive instances in bag-based methods.

- WENO is a plug-and-play framework that can be conveniently applied to any existing bag-based methods using attention mechanism to improve their performance of both instance and bag classification. Extensive experiments on five datasets show that the current SOTA methods ABMIL [13] and DSMIL [18] achieve significant improvements in both instance and bag classification performance when they are combined with WENO. Code will be publicly available.

## 2 Related Work

### 2.1 Deep Multiple Instance Learning

**Instance-based methods** Instance-based methods focus on training an instance classifier and then aggregating the instance predictions in a bag to make the bag prediction. For example, Campanella et al. [2] and Chikontwe et al. [5] iteratively trained the instance classifier by selecting key instances based on the predicted probability of the instance classifier in each iteration and assigning them pseudo labels of the corresponding bags. After that, the former used Recurrent Neural Network (RNN) to aggregate instance predictions to perform bag classification, and the latter proposed a soft-assignment strategy for bag inference.

**Bag-based methods** Bag-based methods focus on aggregating instance features in a bag into a bag feature, and then training a bag classifier with bag labels. Most of them utilize attention mechanism to aggregate instance features, in which Ilse et al. [13] is the first work of this kind. Later, Hashimoto et al. [9] used the attention mechanism to aggregate instance features at different resolutions. Yao, Zhu et al. [40, 38] proposed to first cluster the instances in each bag and then use the attention mechanism to aggregate the features of different clusters. Shi et al. [31] added the instance-level loss to the bag-level loss of [13], but the pseudo labels of each instance still came from its corresponding bag label. Recently, Li et al. [18] proposed to use non-local attention to aggregate instance features. Shao et al. [29], Li et al. [19] proposed to use Transformer to aggregate instance features.

Different from previous MIL methods, we construct a weakly supervised knowledge distillation framework to combine the training of a bag classifier and an instance classifier and utilize the knowledge transfer between the two branches of the distillation framework to mutually enhance both classifiers.

## 2.2 Knowledge Distillation

Knowledge distillation is originally a method to transfer knowledge from a pre-trained complex teacher network to a simpler student network. In the deployment phase, the student network can replace the teacher network to achieve model compression [25, 24, 21, 6, 11]. A common method of distilling knowledge from the teacher network to the student network is to use the output logits of the teacher network as the soft labels for training the student network [12].

Recently, self-distillation techniques are developed to train a student network without a pre-trained teacher network [17, 36, 39, 3]. Instead, the teacher usually has the same structure as the student and is updated by momentum-based moving average of the student network [3]. Figure 1 (a) and (b) concisely illustrates the two main existing knowledge distillation paradigms.

Knowledge distillation is a fast-developing area and we refer to [8] for a comprehensive survey. In this paper, we propose the concept of weakly supervised knowledge distillation for the first time.

# 3 Method

## 3.1 Problem Formulation

### 3.1.1 Multiple Instance Learning (MIL)

Given a dataset $W = \{W_1, W_2, \ldots, W_N\}$ consisting of $N$ WSIs, each WSI $W_i$ is tiled into non-overlapping small patches $\{p_{i,j}, j = 1, 2, \ldots, n_i\}$, where $n_i$ is the number of patches cut out of $W_i$. All patches $p_{i,j}$ in $W_i$ form a bag, where each patch is an instance. The bag label $Y_i \in \{0, 1\}$, $i = \{1, 2, \ldots N\}$ and the instance labels $\{y_{i,j}, j = 1, 2, \ldots, n_i\}$ have the following relationship:

$$Y_i = \begin{cases} 0, & \text{if } \sum_j y_{i,j} = 0 \\ 1, & \text{else} \end{cases} \tag{1}$$

That is, all instances in negative bags are negative, while at least one positive instance exists in a positive bag. In MIL, only the labels of the training bags are available, while the labels of the instances in each positive bag are unknown. Our objective is to accurately predict both the labels of each bag in the test set (bag classification) and the labels of each instance in them (instance classification).

### 3.1.2 Bag-based MIL Methods

We first briefly review bag-based methods for easier understanding of our proposed framework. These methods first use an encoder $f$ to extract features $z_{i,j}$ for all instances $\{p_{i,j}, j = 1, 2, \ldots, n_i\}$ in bag $W_i$, and then aggregate these instance features using a permutation invariant function $g$ to obtain the bag feature $Z_i$:

$$Z_i = g\left(f\left(p_{i,1}\right), f\left(p_{i,2}\right) \ldots\right) \tag{2}$$

Finally, a bag classifier $\varphi$ is utilized to predict the class of the bag.

$$\widehat{Y}_i = \varphi\left(Z_i\right) \tag{3}$$

Traditional aggregation functions $g$ include max-pooling and mean-pooling, while ABMIL [13] presents an attention-based trainable aggregation function:

$$Z_i = \sum_{j=1}^{n_i} a_{i,j} f\left(p_{i,j}\right) \tag{4}$$

$$a_{i,j} = \frac{\exp\left\{w^\top \tanh\left(V h_{i,j}^\top\right)\right\}}{\sum_{j=1}^{n_i} \exp\left\{w^\top \tanh\left(V h_{i,j}^\top\right)\right\}} \tag{5}$$

where $a_{i,j}$ is the attention score predicted by the self-attention network which is parameterized by $w$ and $V$. The subsequent bag-based methods almost all adopt the attention-based aggregation methods, and the difference lies in how to construct the attention weight $a_{i,j}$. The weights $a_{i,j}$ reflect the

contribution of each instance in making the bag prediction, and they can be normalized as the instance prediction for positive bags.

$$\hat{y}_{i,j} = \text{norm}\,(a_{i,j}) \tag{6}$$

The normalization function can be formulated as follows:

$$X_{\text{norm}} = \frac{X - X_{\text{min}}}{X_{\text{max}} - X_{\text{min}}} \tag{7}$$

where $X_n orm$ is the normalized data, $X$ is the original data, and $X_m ax$ and $X_m in$ are the maximum and minimum values of the original data, respectively.

## 3.2   Weakly Supervised Knowledge Distillation Framework

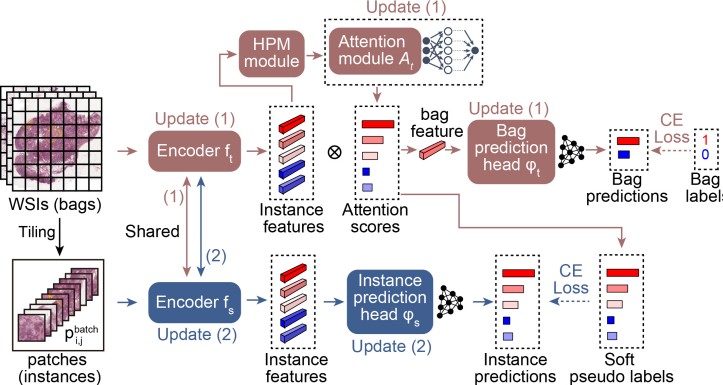

Figure 2: Architecture of the proposed **we**akly supervised k**no**wledge distillation (WENO) framework. WENO contains a teacher branch (the red branch) and a student branch (the blue branch). The teacher branch is essentially a bag-level classifier, which consists of an instance encoder, a hard positive instance mining (HPM) module, an attention module and a bag prediction head. The student branch is essentially an instance-level classifier, which consists of an instance encoder and an instance prediction head. The two branches share the same instance encoder. The teacher branch is trained with bag labels, while the student branch is trained with the attention scores of the teacher branch as soft pseudo labels for the instances in positive bags. Furthermore, we propose a hard positive instance mining strategy (the HPM Module) to leverage the knowledge learned by the student to help the teacher learn hard positive instances. Note that the teacher branch and the student branch are optimized alternately, where the optimization of the teacher branch is represented by update (1) in the figure and the optimization of the student branch is represented by update (2) in the figure.

### 3.2.1   Framework Overview

Figure 2 illustrates the overall framework of our proposed WENO, which contains a teacher branch (the red branch) and a student branch (the blue branch). The whole teacher branch is a bag classifier, which consists of an instance encoder, a hard positive instance mining (HPM) module, an attention module and a bag prediction head. The whole student branch is an instance classifier, which consists of an instance encoder and an instance prediction head. The encoders in the two branches share the same parameters. We directly train the bag classifier in the teacher branch with bag labels. The instance classifier in the student branch is trained with two sets of instances, in which the first set is negative instances from negative bags and the second set is the instances from positive bags with pseudo labels generated according to the normalized attention scores output by the bag classifier in the teacher branch.

Different from the traditional distillation methods in which the teacher is pre-trained or updated by momentum updates from the student, our teacher and student models are trained alternately, so that not only the teacher can transfer the knowledge learned from the bag labels to the student, but also the knowledge learned by the student can be transferred to the teacher through the shared encoder. Furthermore, we propose the hard positive instance mining strategy (the HPM Module) to help the teacher better explore hard positive instances using the knowledge learned by the student, and further improve the network's generalization ability for bag classification.

### 3.2.2 Teacher Network

The teacher branch is a typical bag classifier and one bag is input to it at a time. The instance features $z_{i,j}$ are first extracted using the encoder $f_t$ for all instances $\{p_{i,j}, j = 1, 2, \ldots, n_i\}$ within the bag $W_i$, and then filtered by the hard positive instance mining (HPM) module (See Section 3.3 for details), and then the attention score $a_{i,j}$ for each instance $z_{i,j}$ is obtained by the attention module $A_t$. Finally, the bag feature $Z_i$ is obtained by aggregating the instance features within the bag using attention scores, and input to the bag prediction head $\varphi_t$ to obtain the bag-level prediction $\widehat{Y}_i$. Since the bag label $Y_i$ is available, the teacher branch can be trained in an end-to-end manner.

$$z_{i,j} = f_t\left(p_{i,j}\right) \tag{8}$$

$$a_{i,j} = A_t\left(z_{i,j} \mid z_{i,1}, z_{i,2}, \ldots, z_{i,n_i}\right) \tag{9}$$

$$Z_i = \sum_{j=1}^{n_i} a_{i,j} z_{i,j} \tag{10}$$

$$\widehat{Y}_i = \varphi_t\left(Z_i\right) \tag{11}$$

$$\text{Loss}_{\text{teacher}} = CE\left(Y_i, \widehat{Y}_i\right) \tag{12}$$

The purpose of training the bag-level classifier in the teacher branch is to obtain the attention scores of each instance and use them as soft pseudo labels to train the instance-level classifier in the student branch. Note that the teacher branch can be implemented with any existing bag-based MIL methods using attention mechanism, such as ABMIL [13], DSMIL [18], and TransMIL [29], etc., and we compare the performance of using ABMIL and DSMIL as the teachers in experiments (Section 4.5 and 4.6).

### 3.2.3 Student Network

The student branch is an instance-level classifier which consists of an encoder $f_s$ shared with the teacher branch and an instance prediction head $\varphi_s$. We train the student branch using both the instances in positive bags with the normalized attention scores of the teacher classifier as soft pseudo labels and the instances in negative bags with true negative labels. Different from the teacher branch, the inputs to the student branch are randomly selected instances, which may come from the same bags or different bags.

The loss function of the student branch is the cross-entropy between the network prediction $\hat{y}_{i,j}$ and the label $y_{i,j}$:

$$y_{i,j} = \begin{cases} \text{norm}\left(a_{i,j}\right), & \text{if } Y_i = 1 \\ 0, & \text{else} \end{cases} \tag{13}$$

$$\hat{y}_{i,j} = \varphi_s\left(z_{i,j}\right) \tag{14}$$

$$\text{Loss}_{\text{student}} = CE\left(y_{i,j}, \hat{y}_{i,j}\right) \tag{15}$$

Since the student not only learns from the teacher's attention scores, but also learns from the true negative instances directly, the instance classification ability of the student can surpass the attention scores of the teacher, which is supported by our experimental results (Section 4.5, 4.6 and 5). Moreover, since the student and the teacher share the parameters of the instance encoders, the knowledge learned by the student can also improve the instance-level classification ability of the teacher, which is also validated in the ablation study (Section 5).

In inference, we use the instance classifier in the student branch to make predictions for all instances in a bag, and then use a simple max-pooling to aggregate the predictions of instances to accomplish bag prediction.

### 3.3 Hard Positive Instance Mining Strategy (HPM)

Positive bags contain multiple positive instances, but they differ significantly in the difficulty of identification (e.g., in cancer detection, some patches contain a large number of cancer cells, while some other patches contain a small number of cancer cells). The loss of the bag-based methods is defined at the bag-level, so it only needs to identify at least one positive instance to classify the positive bag correctly. This makes bag-level classifiers tend to learn only easy positive instances but ignore the hard ones during training, which limits both the bag and the instance classification performance. To address this problem, we propose the hard positive instance mining strategy to force the teacher network to continuously learn and mine hard positive instances in the positive bags.

Specifically, we first train the teacher and the student models for a certain number of epochs, after which the student network has a certain instance classification capability. Then, before continuing to train the teacher, we use the student classifier to predict all the instances in the input positive bags, and drop some instances for which the student outputs high positive prediction probability to construct hard pseudo bags. In this way, we can force the bag classifier to keep mining hard positive instances in positive bags to achieve better performance.

## 4 Experiments and Results

### 4.1 Datasets

We used five datasets to comprehensively evaluate the performance of WENO, including two synthetic datasets and three real-world datasets. To explore the performance of WENO under different positive instance ratios, we used the 10-class natural image dataset CIFAR 10 [15] and the 9-class pathological image dataset CRC [14] to construct synthetic WSI datasets with different positive instance ratios, and they are denoted CIFAR-10-MIL dataset and the CRC-MIL dataset, respectively. Furthermore, we used real-world pathology datasets from three different medical centers to evaluate the performance of WENO, including a breast cancer lymph node metastasis public dataset, the Camelyon16 dataset [1], a lung cancer diagnosis public dataset, the TCGA Lung Cancer dataset, and an in-house cervical cancer lymph node metastasis dataset, the Clinical Cervical dataset. Detailed descriptions of the datasets are available in the supplementary material.

### 4.2 Evaluation Metrics

For both instance and bag classification, we use the Area Under Curve (AUC) as the evaluation metric. We report the AUC metrics for the instance and bag classification on two synthetic datasets and the Camelyon 16 dataset. However, since the instance-level ground truth labels for the TCGA Lung Cancer dataset and the Clinical Cervical dataset are not available, we only report their AUC for bag classification.

### 4.3 Implementation Details

For the CIFAR-10-MIL dataset, the encoder in Figure 2 is implemented using the AlexNet [16]. For the other datasets, the encoder is implemented using the ResNet18 [10]. Both the prediction heads and the attention module are implemented using fully connected layers. No pre-training of the network parameters and no image augmentation are performed. The SGD optimizer is used to optimize the network parameters with a fixed learning rate of 0.001. For the hard positive instance mining strategy, we drop the instances with positive probability higher than a threshold in positive bags. The hyperparameter thresholds vary for each dataset, and we used grid search on the validation set to determine the optimal values. In the supplementary material, we give a robustness study of the threshold on the Camelyon 16 dataset. All experiments were performed using 4 Nvidia 3090 GPUs.

### 4.4 Comparison Methods

We compare WENO with a series of state-of-the-art methods. For both the synthetic datasets and the real-world datasets, we use the classic ABMIL [13] and the latest DSMIL [18] as the teachers to construct the WENO frameworks and compare their instance and bag classification performance with SOTA methods. The comparison methods include instance-based methods: RNN-MIL [2] and

Chi-MIL [5]; bag-based methods: ABMIL [13], Loss-ABMIL [31] and DSMIL [18]. We reproduced these methods based on the published codes, and the specific parameter settings are provided in the supplementary material. We also compare the results of using the fully supervised approaches on the synthetic datasets and the Camelyon 16 dataset, i.e., performing supervised training using the true labels of each instance and aggregating the instance predictions using max-pooling to obtain the bag predictions.

## 4.5  Results on Synthetic Datasets

Table 1 and Table 2 show the instance and bag classification performance of WENO on the CIFAR-10-MIL dataset and the CRC-MIL dataset with different positive instance ratios, respectively. We use the classic ABMIL [13] and the latest DSMIL [18] as the teachers to construct the WENO frameworks. It can be seen that the WENO framework significantly improves the performance of the two original bag-based methods for both the bag classification and the instance classification tasks under all positive instance ratios. The advantage of WENO is especially significant for instance classification. In particular, DSMIL [18] does not work well in instance classification under low positive instance ratios, while the performance of DSMIL+WENO is much higher. For bag classification, as shown in Table 1 (b), DSMIL [18] does not work at positive instance ratios of 5% and 10%, while the bag classification AUC after combining the WENO framework reaches 0.9367 and 0.9900, respectively. These results show the powerful advantages of WENO: significant performance gains and easy plug-and-play ability.

Table 1: Results on the CIFAR-10-MIL dataset.

(a) Instance classification AUC.

| Positive patch ratio | 1% | 5% | 10% | 20% | 50% | 70% |
|---|---|---|---|---|---|---|
| Fully supervised | 0.9215 | 0.9621 | 0.9723 | 0.9740 | 0.9699 | 0.9715 |
| ABMIL [13] | 0.6253 | 0.9083 | 0.9241 | 0.9237 | 0.8224 | 0.7935 |
| ABMIL + WENO | **0.7427** | **0.9289** | **0.9492** | **0.9581** | **0.9495** | **0.9454** |
| △ | **+0.1174** | **+0.0206** | **+0.0251** | **+0.0344** | **+0.1271** | **+0.1519** |
| DSMIL [18] | 0.4039 | 0.5515 | 0.4918 | 0.8258 | 0.6152 | 0.7525 |
| DSMIL + WENO | **0.7291** | **0.9408** | **0.9179** | **0.9657** | **0.9393** | **0.9525** |
| △ | **+0.3252** | **+0.3893** | **+0.4261** | **+0.1399** | **+0.3241** | **+0.2000** |

(b) Bag classification AUC.

| Positive patch ratio | 1% | 5% | 10% | 20% | 50% | 70% |
|---|---|---|---|---|---|---|
| Fully supervised | 0.5758 | 0.9531 | 0.9905 | 0.9972 | 1.000 | 1.000 |
| ABMIL [13] | 0.5783 | 0.8850 | 0.9955 | **1.000** | **1.000** | **1.000** |
| ABMIL + WENO | **0.6005** | **0.9300** | **0.9973** | **1.000** | **1.000** | **1.000** |
| △ | **+0.0222** | **+0.0450** | **+0.0018** | - | - | - |
| DSMIL [18] | 0.4025 | 0.5174 | 0.5265 | 0.9468 | 0.9850 | **1.000** |
| DSMIL + WENO | **0.4069** | **0.9367** | **0.9900** | **1.000** | **1.000** | **1.000** |
| △ | **+0.0044** | **+0.4193** | **+0.4635** | **+0.0532** | **+0.0150** | - |

Table 2: Results on the CRC-MIL dataset.

(a) Instance classification AUC.

| Positive patch ratio | 10% | 20% | 50% | 70% |
|---|---|---|---|---|
| Fully supervised | 0.9978 | 0.9976 | 0.9977 | 0.9971 |
| ABMIL [13] | 0.7410 | 0.8729 | 0.8800 | 0.7965 |
| ABMIL + WENO | **0.9625** | **0.9819** | **0.9759** | **0.9786** |
| △ | **+0.2215** | **+0.1090** | **+0.0959** | **+0.1821** |
| DSMIL [18] | 0.3690 | 0.7008 | 0.4835 | 0.7399 |
| DSMIL + WENO | **0.9697** | **0.9801** | **0.9817** | **0.9760** |
| △ | **+0.6007** | **+0.2793** | **+0.4982** | **+0.2361** |

(b) Bag classification AUC.

| Positive patch ratio | 10% | 20% | 50% | 70% |
|---|---|---|---|---|
| Fully supervised | 1.000 | 1.000 | 1.000 | 1.000 |
| ABMIL [13] | 0.9754 | **1.000** | 0.9766 | 0.9894 |
| ABMIL + WENO | **1.0000** | **1.000** | **0.9957** | **1.0000** |
| △ | **+0.0246** | - | **+0.0191** | **+0.0106** |
| DSMIL [18] | **1.000** | 0.8914 | 0.9997 | **1.000** |
| DSMIL + WENO | **1.000** | **1.0000** | **1.0000** | **1.000** |
| △ | - | **+0.1086** | **+0.0003** | - |

## 4.6  Results on Real-World Datasets

Table 3 (a) shows the performance of WENO on the Camelyon16 dataset for the instance and bag classification. We still use ABMIL [13] and DSMIL [18] as the teachers to construct the WENO frameworks. It can be seen that using WENO brings a significant performance improvement in both classification tasks. Notably, with only bag-level weak labels, the best instance classification AUC (0.9377) obtained with WENO is lower than that of fully supervised method (0.9644) by only 0.0267; while the best bag classification AUC (0.8663) exceeds the fully supervised (0.8621) method, showing the powerful performance of WENO again.

Table 3 (b) shows the bag classification performance of WENO on the TCGA Lung Cancer dataset. We use ABMIL [13] and DSMIL [18] as the teachers to construct the WENO frameworks and achieve the best performance.

Table 3 (c) shows the bag classification performance of WENO on the Clinical Cervical dataset. We use ABMIL [13] and DSMIL [18] as the teachers to construct the WENO frameworks and achieve the optimal performance. In contrast to previous clinical tumor recognition and classification tasks,

the prediction of lymph node metastasis according to the WSIs of primary lesion in this dataset is a very challenging task in current clinical practice. Since there is no prior knowledge about what image features of the primary lesion indicate metastasis, even very experienced pathologists are unable to clearly distinguish between positive and negative instances. WENO achieves the best performance among all comparison methods, which on the one hand indicates that WENO can be applied to the prediction of metastasis using primary lesion WSIs in clinical practice, and on the other hand suggests the potential of WENO to detect underlying pathological patterns from high confident instances.

Some examples of positive and negative slides/patches in the three real-world datasets and the visualization of the heatmaps on the Camelyon16 dataset are given in the supplementary material.

Table 3: Results on the three real-world datasets.

(a) Camelyon16 Dataset.

| Methods | Instance-level AUC | Bag-level AUC |
|---|---|---|
| Fully supervised | 0.9644 | 0.8621 |
| Loss-ABMIL [31] | 0.8995 | 0.7965 |
| ChiMIL [5] | 0.7880 | 0.7025 |
| ABMIL [13] | 0.8480 | 0.8379 |
| ABMIL + WENO | 0.9271 | **0.8663** |
| $\Delta$ | **+0.0791** | **+0.0284** |
| DSMIL [18] | 0.8568 | 0.8401 |
| DSMIL+ WENO | **0.9377** | 0.8495 |
| $\Delta$ | **+0.0809** | **+0.0094** |

(b) TCGA Lung Cancer Dataset.

| Methods | Bag-level AUC |
|---|---|
| Mean-pooling | 0.9369 |
| Max-pooling | 0.9014 |
| RNN-MIL [2] | 0.9107 |
| ABMIL [13] | 0.9488 |
| ABMIL + WENO | 0.9663 |
| $\Delta$ | **+0.0175** |
| DSMIL [18] | 0.9633 |
| DSMIL + WENO | **0.9727** |
| $\Delta$ | **+0.0094** |

(c) Clinical Cervical Dataset.

| Methods | Bag-level AUC |
|---|---|
| Loss-ABMIL [31] | 0.5833 |
| ChiMIL [5] | 0.7425 |
| ABMIL [13] | 0.6446 |
| ABMIL + WENO | 0.8056 |
| $\Delta$ | **+0.1610** |
| DSMIL [18] | 0.8022 |
| DSMIL + WENO | **0.8222** |
| $\Delta$ | **+0.0200** |

## 5 Ablation Study

Figure 3 shows the results of the ablation study of WENO on the CIFAR-10-MIL dataset with a positive instance ratio of 0.2. In this experiment, we use the ABMIL [13] as the teacher to construct the WENO framework. By analyzing the curves in Figure 3, we can find: (1) Comparing the curves of the raw 'ABMIL' and 'ABMIL+WENO (without HPM)' in panel (a) and panel (b), it can be seen that when the bag-level AUC reaches the maximum at about the 25th epoch, their instance-level classification ability is still poor. In addition, the instance-level AUC of the raw ABMIL gets worse as the training proceeds, indicating that the network gradually tends to distinguish positive bags by simple positive instances only. As shown in Figure 3 (b), when WENO is used with ABMIL, the instance-level classification capability is significantly improved, and the network is still able to continuously improve the instance classification capability as the training proceeds. 2) Comparing the curves of the 'ABMIL+WENO (without HPM)' and 'ABMIL+WENO (with HPM)' in panel (b) and panel (c), it can be seen that the proposed hard positive instance mining strategy can further improve the instance classification ability of both the student and the teacher by forcing the teacher network to continuously learn and mine hard positive instances in positive bags. HPM is used after the 150th epoch.

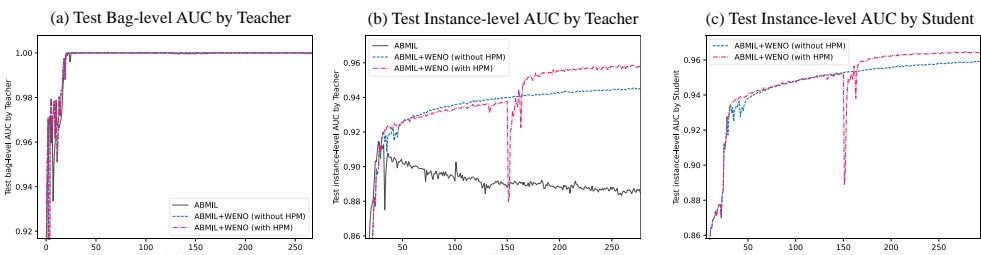

Figure 3: Ablation study curves on the CIFAR-10-MIL dataset.

Table 4 shows the results of the ablation study of key components of WENO on the Camelyon16 dataset. We construct the WENO framework with ABMIL [13] as the teacher, where '**Distillation**' represents whether distillation is used (without 'Distillation' denotes the raw ABMIL), '**Shared Encoder**' represents whether the encoders of the teacher and the student share parameters, and '**HPM**' indicates whether hard instance mining strategy is performed or not. The AUCs of both instance and bag classification indicate the effectiveness of each component of WENO.

Table 4: Ablation study on the Camelyon16 Dataset.

| Distillation | Shared Encoder | HPM | Instance-level AUC | Bag-level AUC |
|:---:|:---:|:---:|:---:|:---:|
| | | | 0.8480 | 0.8379 |
| ✓ | | | 0.8787 | 0.8574 |
| ✓ | ✓ | | 0.9011 | 0.8583 |
| ✓ | ✓ | ✓ | **0.9271** | **0.8663** |

# 6 Visualization of the predictions on the Camelyon16 Dataset

Figure 4 shows some representative predictions on the Camelyon16 dataset. As can be seen, using only slide-level labels, our method is able to accurately predict almost all positive instances, regardless of whether there is a large or small proportion of positive regions in the WSIs. This intuitively demonstrates the powerful performance of WENO and its great potential for clinical applications.

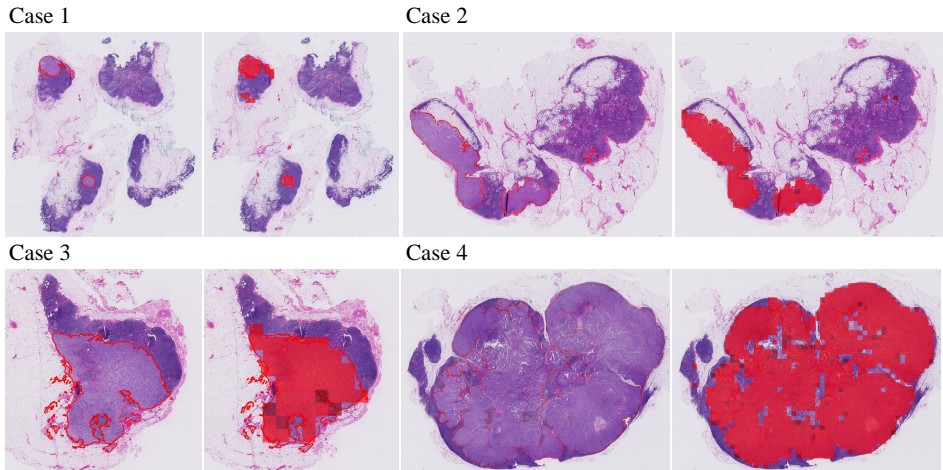

Figure 4: Visualization of the representative predictions on the Camelyon16 dataset, where four cases are included. For each case, the left image is the original image, where the contours of the ground truth positive regions are outlined using red lines and the right image is the predicted image, where the predicted positive instances are marked using red squares and the predicted negative instances are not marked.

# 7 Conclusion

In this paper, we propose WENO, an end-to-end weakly supervised knowledge distillation framework for whole slide image classification. WENO is a plug-and-play framework that can use any existing bag-based MIL methods using attention mechanism as the teacher branch and improves its performance on both bag and instance classification. WENO trains the student classifier with the normalized attention scores of the teacher branch through knowledge distillation, and uses the hard positive instance mining (HPM) strategy to force the teach network to further mine and learn from hard positive instances. In experiments, WENO shows strong instance and bag classification performance on all five datasets, reaching new SOTA. WENO also has the potential to be applied to other MIL problems. As for the limitations, the HPM strategy requires searching for optimal parameters on the validation set, which may prolong the training time. Deep learning-based WSI analysis has a long history, so our study has the same potential negative societal impacts as existing studies.

## Acknowledgments

This work was supported by National Natural Science Foundation of China under Grant 82072021.

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
