# Supplementary Material

**Linhao Qu**[*]
Digital Medical Research Center,
School of Basic Medical Science,
Fudan University
lhqu20@fudan.edu.cn

**Xiaoyuan Luo**[*]
Digital Medical Research Center,
School of Basic Medical Science,
Fudan University
19111010030@fudan.edu.cn

**Manning Wang**[†]
Digital Medical Research Center,
School of Basic Medical Science,
Fudan University
mnwang@fudan.edu.cn

**Zhijian Song**[†]
Digital Medical Research Center,
School of Basic Medical Science,
Fudan University
zjsong@fudan.edu.cn

The supplementary material consists of four sections. Section 1 introduces the details of the five datasets, section 2 shows the robustness study of the threshold of the hard positive instance mining (HPM) strategy on the Camelyon16 dataset, section 3 shows the robustness study of the starting epoch of HPM strategy on the CIFAR-10-MIL dataset and section 4 focuses on the specific parameter settings of the comparison methods.

## 1 Details of the Five Datasets

### 1.1 Synthetic Datasets

To evaluate the performance of WENO under different positive instance ratios and compare it with SOTA algorithms, we used the 10-class natural image dataset CIFAR-10 [6] and the 9-class pathological image dataset CRC-100K [5] to construct WSI datasets with different positive instance ratios, namely CIFAR-10-MIL dataset and CRC-MIL dataset, respectively.

**CIFAR-10-MIL Dataset.** The CIFAR-10 dataset [6] consists of 60000 $32 \times 32$ color images in 10 classes (airplane, automobile, bird, cat, deer, dog, frog, horse, ship, truck), with 6000 images per class. There are 50000 training images and 10000 testing images. To simulate the pathological Whole Slide Image (WSI), we combined a random set of images from each of the 10 categories in the CIFAR-10 dataset to construct a WSI. Specifically, we considered each image of each category in the CIFAR-10 dataset as an instance, where only all instances of the "truck" category were labeled as positive and other instances as negative (the truck category was randomly selected). Then, we randomly selected $a$ positive instances and $100 - a$ negative instances from all instances without replacement to form a positive bag with a positive instance ratio of $\frac{a}{100}$. Similarly, we randomly selected 100 negative instances without replacement to form a negative bag. This process was repeated until all the positive or negative instances in the CIFAR-10 dataset were used up. By adjusting the value of $a$, we constructed six subsets of the CIFAR-10-MIL dataset, and the positive instance ratios of the six subsets were 1%, 5%, 10%, 20%, 50%, and 70%, respectively. The number of bags in the training and testing sets on the CIFAR-10-MIL dataset under different positive instance ratios is shown in Table 1. Figure 1 shows the representative synthetic WSIs of the CIFAR-10-MIL dataset under each positive instance ratio.

---

[*]Equal Contribution.

[†]Corresponding Authors. All authors are also from Shanghai Key Lab of Medical Image Computing and Computer Assisted Intervention.

36th Conference on Neural Information Processing Systems (NeurIPS 2022).

Table 1: The number of bags in the training and testing sets on the CIFAR-10-MIL dataset under different positive instance ratios.

| Positive instance ratio | 1% | 5% | 10% | 20% | 50% | 70% |
|---|---|---|---|---|---|---|
| Num of bags (train: test) | 452:90 | 459:91 | 472:94 | 500:100 | 200:40 | 142:28 |

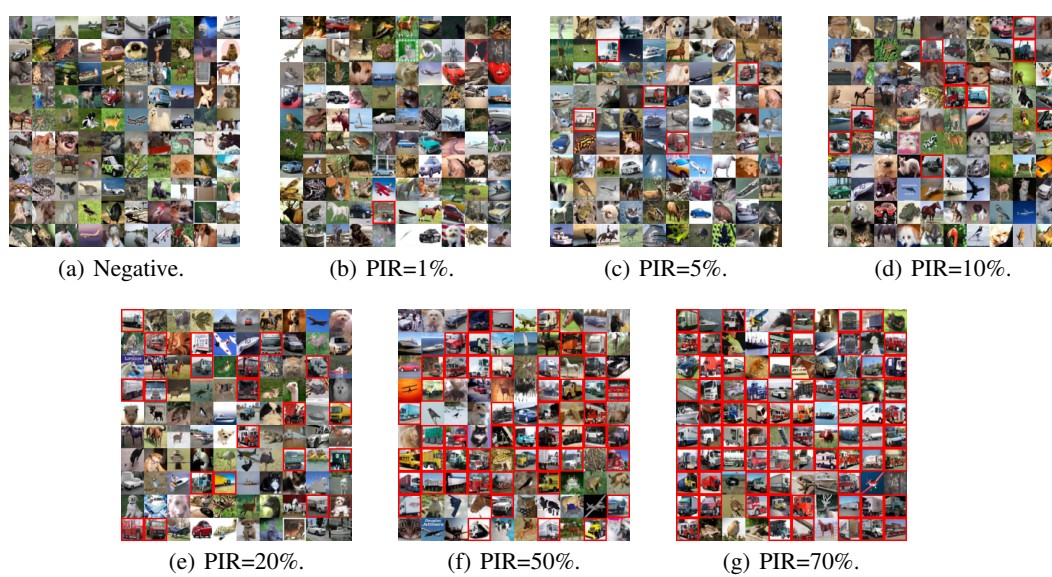

| (a) Negative. | (b) PIR=1%. | (c) PIR=5%. | (d) PIR=10%. |
|---|---|---|---|

| (e) PIR=20%. | (f) PIR=50%. | (g) PIR=70%. |
|---|---|---|

Figure 1: Examples of representative synthetic WSIs of CIFAR-10-MIL dataset, including a negative slide and six positive slides with positive instance ratio (PIR) of 1%, 5%, 10%, 20%, 50% and 70%. The positive instances in each slide are marked by red boxes.

**CRC-MIL Dataset.** The CRC-100K dataset [5] is a real pathology image dataset containing 100,000 images of human colorectal cancer and healthy tissue with detailed category annotation. The dataset is sampled at $20\times$ magnification for $224\times224$ patches and contains the following nine non-overlapping pathology categories, namely adipose, background, debris, lymphocytes, mucus, smooth muscle, normal colon mucosa, cancer-associated stroma, and colorectal adenocarcinoma epithelium. We used the CRC-100K dataset to construct the CRC-MIL dataset with different positive instance ratios. Consistent with the CIFAR-10-MIL dataset, we set all instances in the category "colorectal adenocarcinoma epithelium" as positive instances and the remaining instances as negative instances. We randomly selected $a$ positive instances and $100 - a$ negative instances from all instances without replacement to form a positive bag with a positive instance ratio of $\frac{a}{100}$. Similarly, we randomly selected 100 negative instances without replacement to form a negative bag. This process was repeated until all the positive or negative instances in the CRC-100K dataset were used up. By adjusting the value of $a$, we constructed four subsets of the CRC-MIL dataset and the positive instance ratios of the four subsets were 10%, 20%, 50% and 70%, respectively. The number of bags in the training and testing sets on the CRC-MIL dataset under different positive instance ratios is shown in Table 2. Figure 2 shows some examples of representative synthetic WSIs of the CRC-MIL dataset.

Table 2: The number of bags in the training and testing sets on the CRC-MIL dataset under different positive instance ratios.

| Positive instance ratio | 10% | 20% | 50% | 70% |
|---|---|---|---|---|
| Num of bags (train: test) | 720:180 | 760:190 | 458:114 | 326:80 |

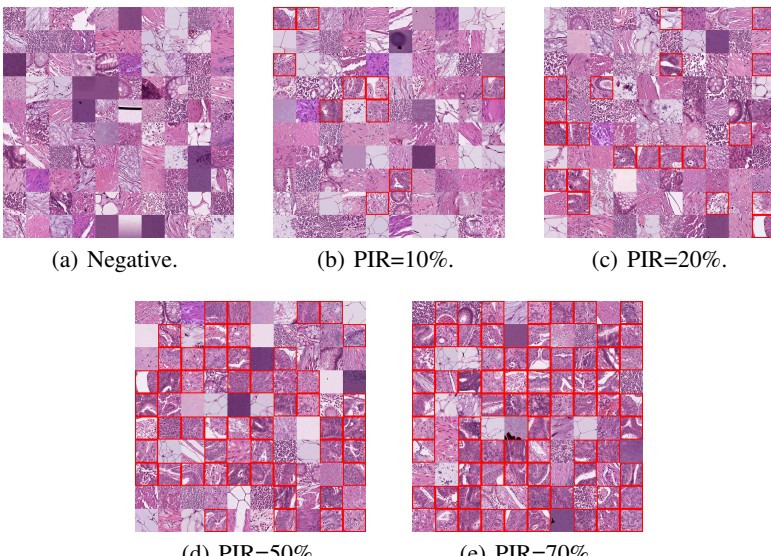

|              |              |              |
|:------------:|:------------:|:------------:|
| (a) Negative. | (b) PIR=10%. | (c) PIR=20%. |
| (d) PIR=50%. | (e) PIR=70%. |              |

Figure 2: Examples of representative synthetic WSIs of CRC-MIL dataset, including a negative slide and four positive slides with positive instance ratio (PIR) of 10%, 20%, 50% and 70%. The positive instances in each slide are marked by red boxes.

## 1.2 Real-World Datasets

We used three real-world datasets from three different centers to evaluate the performance of WENO in real-world pathology diagnosis. The real-world datasets contain a breast cancer lymph node metastasis public dataset, Camelyon16 dataset [1], a lung cancer diagnosis public dataset, TCGA Lung Cancer dataset, and a cervical cancer lymph node metastasis in-house dataset, Clinical Cervical dataset.

**Camelyon16 Dataset.** Camelyon16 dataset is a public histopathology image dataset for detecting breast cancer metastasis in lymph nodes [1], and the dataset contains 400 WSIs of lymph nodes (270 for training and 130 for testing). The WSIs that contained metastasis were labeled positive, and the others were labeled negative. The dataset provides not only the labels of whether a WSI is positive but also pixel-level labels of the metastasis areas. We only used the slide-level labels for training to satisfy the weakly supervised scenarios, and used the pixel-level labels of cancer areas to evaluate the instance classification performance of each algorithm. For preprocessing, we divide each WSI into $512 \times 512$ image patches without overlapping under $10 \times$ magnification. Similar to DSMIL [7], patches with an entropy less than 5 are dropped out as background. A patch is labeled as positive if it contains 25% or more cancer areas; otherwise, it is labeled as negative. Finally, a total of 186,604 instances were obtained, of which there were only 8117 positive instances (4.3%). The number of positive instances in different positive bags varied greatly, and some positive bags contained only several positive instances. Figure 3 provides some representative examples of WSIs in the Camelyon16 dataset.

**TCGA Lung Cancer Dataset.** The TCGA Lung Cancer dataset includes a total of 1054 WSIs from The Cancer Genome Atlas (TCGA) Data Portal, which includes two sub-types of lung cancer, namely Lung Adenocarcinoma and Lung Squamous Cell Carcinoma. Our goal is to accurately classify the two subtypes, where WSIs of Lung Adenocarcinoma are labeled as negative and WSIs of Lung Squamous Cell Carcinoma are labeled as positive. Only slide-level labels are available for this dataset, and patch-level labels are unknown. This dataset contains about 5.2 million patches at $20 \times$ magnification, with an average of about 5,000 patches per bag. DSMIL [7] further mapped these patches to 512-dimensional features using contrastive self-supervised learning SimCLR [2]. Our experimental settings were exactly the same as DSMIL [7] and the features provided by DSMIL were used as input. We randomly divided the features of these WSIs into a training set (a total of 840 features) and a test set (a total of 210 features). Figure 4 shows some representative examples of WSIs in the TCGA Lung Cancer dataset.

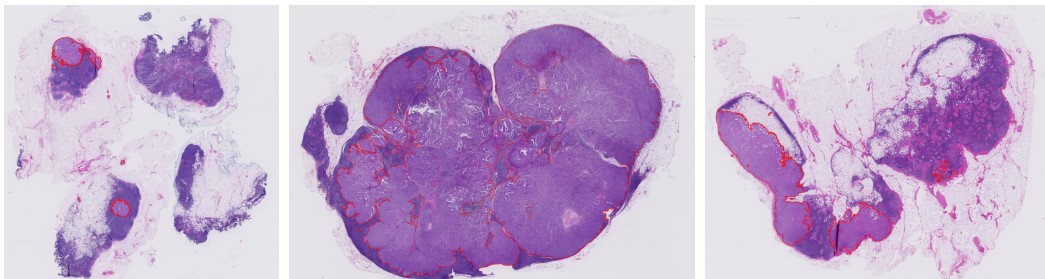

Figure 3: Visualization of some representative examples of WSIs in the Camelyon16 dataset. The images are original positive WSIs with different sizes of positive areas, which are outlined by red lines.

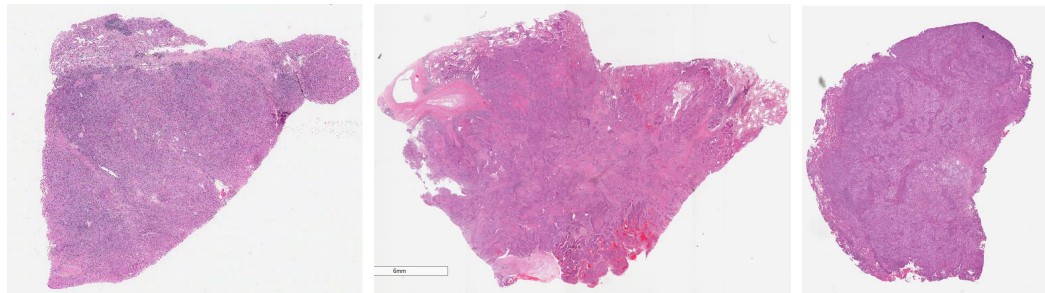

Figure 4: Visualization of some representative examples of WSIs in the TCGA Lung Cancer dataset.

**Clinical Cervical Dataset.** Clinical Cervical Dataset is an inhouse cervical cancer primary lesion pathology dataset that is used to directly predict a patient's pelvic (including para-aortic) lymph node metastasis status from their primary lesion pathology slides. This is a pressing clinical challenge because of its clinical importance for the choice of treatment modality for patients. Unlike the two previous real-world datasets, this task is particularly difficult because there is no prior knowledge about what image features of the primary lesion indicate metastasis, even very experienced pathologists are unable to clearly distinguish between positive and negative instances. This dataset includes a total of 374 WSIs from different patients with cervical cancer, among which those with pelvic lymph node metastasis are labeled as positive (209 cases) and those without pelvic lymph node metastasis are labeled as negative (165 cases). We randomly divided the WSIs into a training set (300 cases) and a test set (74 cases) with an approximate ratio of 4:1, and the WSI was pre-processed in a similar way to Camelyon16, including removing the background regions and dividing the slides into $224 \times 224$ patches at $10\times$ magnification. Finally, 297,831 patches are obtained after preprocessing. Figure 5 provides some representative examples of WSIs in the Clinical Cervical Dataset.

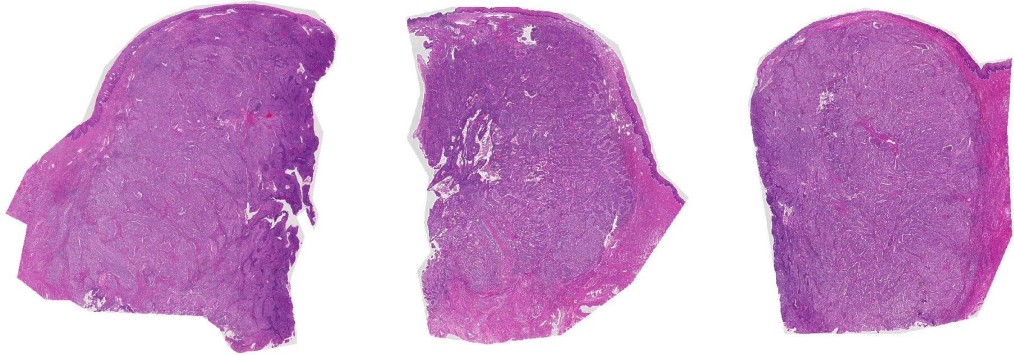

Figure 5: Visualization of some representative examples of WSIs in the Clinical Cervical dataset.

## 2 Robustness Study of the Threshold of the Hard Positive Instance Mining Strategy on the Camelyon16 Dataset

We propose the hard positive instance mining (HPM) strategy to force the teacher network to continuously learn and mine hard positive instances in the positive bags. Specifically, we first train the teacher and the student models for a certain number of epochs, after which the student network has a certain instance classification capability. Then, before continuing to train the teacher, we use the student classifier to predict all the instances in the input positive bags, and drop the instances with positive probability higher than a threshold in positive bags to construct hard pseudo bags. We study the robustness of WENO to this threshold on the Camelyon16 dataset, and the results are shown in Table 3. We used the ABMIL [4] to construct the WENO framework. In the table, "**w/o HPM**" means that HPM strategy is not used and "**w/ HPM**" means that HPM strategy is used. When threshold=1.00, it is equivalent to not using the HPM strategy. It can be seen that the instance-level AUC with HPM strategy is significantly higher than that without HPM strategy at all thresholds; when the threshold is below 0.92, the bag-level AUC with HPM strategy is slightly lower than without HPM strategy, which may be caused by dropping too many positive instances. The instance-level AUC and bag-level AUC using HPM strategy are higher than the raw ABMIL at all thresholds. It can be seen that the optimal threshold is actually a trade-off between instance-level AUC and bag-level AUC, and we finally selected threshold=0.94 as the optimal threshold. The above robustness study also shows that the HPM strategy has a strong robustness.

Table 3: Robustness study of the threshold on the Camelyon16 dataset.

| Method | Threshold | Instance-level AUC | Bag-level AUC |
|---|---|---|---|
| ABMIL [4] | - | 0.8480 | 0.8379 |
| ABMIL+WENO (w/o HPM) | 1.0000 | 0.8787 | 0.8574 |
| ABMIL+WENO (w/ HPM) | 0.9800 | 0.8884 | **0.8717** |
| ABMIL+WENO (w/ HPM) | 0.9600 | 0.9196 | 0.8675 |
| ABMIL+WENO (w/ HPM) | 0.9400 | **0.9271** | 0.8663 |
| ABMIL+WENO (w/ HPM) | 0.9200 | 0.9256 | 0.8402 |
| ABMIL+WENO (w/ HPM) | 0.9000 | 0.9204 | 0.8495 |
| ABMIL+WENO (w/ HPM) | 0.8800 | 0.9042 | 0.8478 |

## 3 Robustness Study of the Starting Epoch of HPM Strategy on the CIFAR-10-MIL Dataset

We performed a robustness study on the starting epoch of the HPM strategy, and the results are shown in Figure 6. We construct the WENO framework using ABMIL [4] as the teacher and experimented on the CIFAR-10-MIL dataset with a positive instance ratio of 0.2. In the Figure, panel (a) represents the AUC metric for bag-level classification using the teacher network, panel (b) represents the AUC metric for instance-level classification using the normalized attention scores of the teacher network, and panel (c) represents the AUC metric for instance-level classification using the student network. The gray line (WENO + 100 epoch HPM) indicates that the HPM module is introduced at the 100th epoch, the blue line (WENO + 150 epoch HPM) indicates that the HPM module is introduced at the 150th epoch, and the red line (WENO + 200 epoch HPM) indicates that the HPM module is introduced at the 200th epoch. Comparing the gray, blue and red lines in panel (a), panel (b) and panel (c), it can be seen that there is little difference in the final performance regarding the bag-level classification and instance-level classification, whether HPM is introduced at the 100th, 150th and 200th epochs. In this experiment, we added HPM strategy at three different time point, the 100th, 150th and 200th epoch, and the results show that their final instance-level classification AUC of are almost the same for both the teacher and the student network. These results indicate that HPM is not sensitive to the starting epoch.

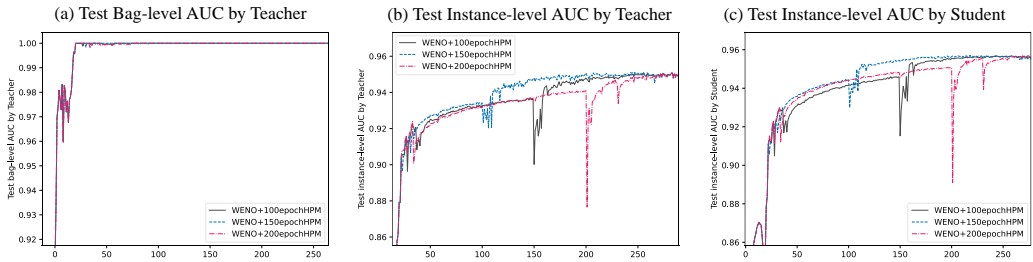

Figure 6: Robustness Study of the Starting Epoch of the HPM Strategy on the CIFAR-10-MIL dataset.

# 4 Specific Parameter Settings of the Comparison Methods

For all datasets except the TCGA Lung Cancer dataset, we performed a grid search on the key hyperparameters in all the comparison methods. For all comparison methods, we search learning rate over {0.0001, 0.0005, 0.001, 0.005, 0.01}. For Chi-MIL [3], we also search hyperparameter k over grid {2, 4, 8, 16, 32, 64}. For Loss-ABMIL [8], we search hyperparameter $\lambda$ over grid {0.01, 0.1, 1, 10}. For the TCGA Lung Cancer dataset, since our experimental settings and inputs are exactly the same as those of DSMIL [7], we directly report the experimental results and comparison methods of DSMIL [7].