# OpenReview forum: "Bi-directional Weakly Supervised Knowledge Distillation for Whole Slide Image Classification"
_NeurIPS.cc/2022/Conference — NeurIPS 2022 Accept_

### Official Review · Reviewer_hPPu · 2022-07-08

**Rating:** 8
**Confidence:** 5
**Soundness:** 4 excellent
**Presentation:** 4 excellent
**Contribution:** 3 good

**Summary:**

This paper presents a novel method to train a teacher-student attention-MIL model with weak supervision in a way that the student can be used to improve the teacher. Dubbed WENO, the method shares an encoder between the teacher and student model and while the teacher's bag-level attention scores are used to train the student (distillation), the student's instance model is used to select hard examples for the teacher's bag. This is designed to avoid a common pitfall of MIL models where the teacher tends to use only easy-to-classify positive instances from the bag and becomes weak at classifying hard examples. The method can be added to existing attention-based MIL methods. The method is evaluated on 2 synthetic and 3 real-world datasets of WSI histology images, and is using 2 SOTA MIL methods (ABMIL and DSMIL). The method is shown to outperform SOTA by a significant margin on datasets where the prevalence of the positive class is low, and by a small margin for real-world datasets. An ablation study shows that that each key part of the method (distillation, shared-encoder and hard-example-selector) all participate in improving the baseline.


**Questions:**

no questions.

**Limitations:**

Beyond the author-stated limitation related to hyperparameter tuning, i see no further limitations of the approach.


**Strengths And Weaknesses:**

strengths:
- the method combines for the first time multi-instance-learning (MIL) with weakly-supervised knowledge distillation.
- the method addresses the issue with attention-based MIL methods where the teacher's performance stops improving after a while and even decreases because it focuses more and more on a few easy-to-classify examples. The proposed HPM (hard-positive-mining) module to alleviate this issue is novel and uses the student's predictions after a set number of epochs to select hard examples for the teacher's bag.
- the method improves upon SOTA for 3 public datasets of WSI histology. It comes ever-closer (within 3%) to fully-supervised models trained with instance-level labels (expensive to obtain).
- the ablation study is useful and convincing.
- the method is not specific to WSI histology images and can be applied to any MIL setting
- the method does not affect inference complexity, only training is affected. This is a key factor in most real-world applications.
- the paper is very well written and clarity is excellent

weaknesses:
- the achieved improvement over SOTA is relatively small for the real-world datasets.
- hyperparameter tuning has to be performed for the HPM module to determine the best threshold. This is discussed in the supplemental material. However, another hyperparameter, the number of epochs after which to bring the HPM module online is not discussed.

---

> ### Author Response · Authors · 2022-08-02
> **Response to Reviewer hPPu**
>
> Thank you very much for your valuable comments, which are very helpful to further improve our manuscript. We will respond to the weaknesses in detail. Some figures and results of newly added experiments can be found in Page 6, Section Rebuttal in the newly submitted ‘Supplementary Material.pdf' file.
>
> ### - Weaknesses
>
> > 1. the achieved improvement over SOTA is relatively small for the real-world datasets.
>
> * **Response**：
>
>   In this paper, we used three real-world datasets (Camelyon16 Dataset, TCGA Lung Cancer Dataset and Clinical Cervical Dataset) to evaluate the performance of our framework. On the Camelyon16 Dataset, WENO improves the instance-level AUC significantly by **7~8 percentage points**; although the improvement for the bag-level AUC is relatively small (**1~2 percentage points)**. For the TCGA Lung Cancer Dataset and Clinical Cervical Dataset, we could not validate the instance-level metrics because there is no instance-level annotation. Our method still improves the bag-level AUC by **1~2 percentage points** on TCGA Lung Cancer Dataset, although the performance of existing methods is already very high and the room for improvement is very small. Our method also improves bag-level AUC by **1~2 percentage points** on Clinical Cervical Dataset.
>
> > 2. hyperparameter tuning has to be performed for the HPM module to determine the best threshold. This is discussed in the supplemental material. However, another hyperparameter, the number of epochs after which to bring the HPM module online is not discussed.
>
> * **Response**：
>
>   We added a new robustness study on the starting epoch of the HPM strategy, and the results are shown in Page 6 Section 5 (Rebuttal) Figure 7 of the newly submitted ‘Supplementary Material.pdf’ file. We construct the WENO framework using ABMIL [1] as the teacher and experimented on a CIFAR-10-MIL dataset with a positive instance ratio of 0.2. In this experiment, we added the HPM strategy at three different time points (the 100th, 150th and 200th epoch) and the results show that their final instance-level AUCs are almost the same for both the teacher and the student networks. These results indicate that **HPM is not sensitive to the starting epoch.**
>
>
>
>  [1] Ilse et al. "Attention-based deep multiple instance learning." ICML, 2018.

---

> ### Comment · Reviewer_hPPu · 2022-08-08
> **acknowledgment of author's rebuttal**
>
> I appreciate the author's rebuttal and response to the questions. This helps confirm my good score.

---

### Official Review · Reviewer_oxi9 · 2022-07-11

**Rating:** 6
**Confidence:** 5
**Soundness:** 1 poor
**Presentation:** 2 fair
**Contribution:** 2 fair

**Summary:**

This work presents a method to improve instance classification in whole slide images (WSI) using only class labels of WSI. To this end, the authors consider a knowledge distillation method where a teacher and a student model learn from each other alternatively. The teacher model is a bag (WSI) classifier while the student model is an instance classifier. The knowledge transfer between the two models is instance score attention which indicates the contribution of the instance to the bag prediction. Experimental validation was performed over 2 synthetic datasets and 3 real-world datasets for histology data, and the proposed method outperforms several SOTA methods.

**Questions:**

Can you please clarify what is the 'norm' in Equation 6?

**Limitations:**

Please refer to the weaknesses described in Strengths And Weaknesses.

**Strengths And Weaknesses:**

 Strengths:
+ The paper is clearly written and organized. Their problem statement and the proposed approach are clearly described.
+ The paper discusses an important application that is improving the instance classification in WSI data.
+ The empirical results show the benefit of the method, and some ablation studies are provided.

Weaknesses:
 - On page 2, lines 63-66, the authors mentioned that using soft labels as they did helps to alleviate the noisy pseudo-labels used in other methods. It is not clear how it is the case. From the description of the method, the pseudo-labels in this work change every SGD step since the network that builds them changes constantly. Therefore, they are noisier, especially at the beginning of the training. It is not clear why the authors claimed that their pseudo-labels alleviate the noise issue. In addition, there is no empirical evidence to support this claim.
 - On page 2, lines 70-72, the authors mentioned that sharing the feature extractor between student and teacher enhances knowledge distillation. It is not clear why it is the case.
 - In the entire paper, the pseudo-labels were described as the attention scores. It is not until eq.12 (page 6) that the authors describe the soft labels as being single scores that simulate probability. The text should be clear about this detail from the outset. In addition to fig.2,  there should be an operation after attention scores come out from the teacher, before being used by the student’s CE loss.
 - Hard positive instance mining (hpm) proposed in this paper is similar to hard mining in class-activation maps (CAMs). In the latter, strong activations are suppressed, thereby pushing the network to seek more positive regions. The same here, the authors drop high positive instances simulating suppression. Therefore, the proposed mining strategy is not really new; and literature discussion is necessary. See an example for CAMs ib Wei, Y., Feng, J., Liang, X., Cheng, M.-M., Zhao, Y., and Yan, S. (2017). Object region mining with adversarial erasing: A simple classification to semantic segmentation approach. CVPR 2017.
- The main issue in this work is that it is unclear why there is an improvement? The authors are not using additional supervision. In fact, the pseudo labels are expected to be noisier and change constantly. It is unclear why training a student in parallel helps instance classification even though there are no instance labels. Authors are encouraged to provide further analysis to explain the observed improvement. If I had to guess, the main element that improves performance is in eq.12, in particular the zero-label of instances in negative bags. Authors explicitly provided instance labels but only for negative instances, giving their model a large advantage. Knowing what is not positive helps the model.
- Finally, there is an important element missing in the experimental section: the classification accuracy per type of instance (positive/negative, ...). We do not know what pushed the AUC at instance-level to improve. Is it improving the positive instances or negative instances? My guess is the latter helps more since labels of negative instances are explicitly used.
-  The code was not made available. Although the method is well described, there is a concern that the results in this paper would be difficult for a reader to reproduce.

---

> ### Author Response · Authors · 2022-08-02
> **Response to Reviewer oxi9 (Part 6/6)**
>
> > 6. Finally, there is an important element missing in the experimental section: the classification accuracy per type of instance (positive/negative, ...). We do not know what pushed the AUC at instance-level to improve. Is it improving the positive instances or negative instances? My guess is the latter helps more since labels of negative instances are explicitly used.
>
> * **Response**：
>
>   In pathological images, the problem of imbalance between the number of positive instances and negative instances typically exists. For example, the positive rate of patches in the Camelyon16 Dataset is only 4.3%, while that in the TCGA Dataset is over 80%. Therefore, we adopt the AUC metric to evaluate the performance of the model, which is a common evaluation metric for similar studies.
>
>   In addition, we added additional metrics on CIFAR-10-MIL with a positive instance ratio of 0.2, which included True Positive Rate (TPR), True Negative Rate (TNR), False Positive Rate (FPR), False Negative Rate, Accuracy (with threshold=0.5) and AUC. The results are shown in the following Table. In the Table, the first row indicates the metrics of the pseudo labels when the instances in positive bags inherit the positive labels directly; the second row indicates the metrics of our framework with ABMIL as the teacher.
>
>   It can be seen that despite the absence of true positive labels, WENO generates soft pseudo labels with an accuracy of 0.8826, while the method of directly inheriting bag labels has an accuracy of only 20%.
>
>   |                       |  TPR   |  TNR   |  FPR   |  FNR   | Accuracy |  AUC   |
>   | :-------------------: | :----: | :----: | :----: | :----: | :------: | :----: |
>   | inheriting bag labels | 1.0000 | 0.0000 | 1.0000 | 0.0000 |  0.2000  | 0.5000 |
>   |     ABMIL + WENO      | 0.4402 | 0.9932 | 0.0068 | 0.5598 |  0.8826  | 0.9308 |
>
>
>
> > 7. The code was not made available. Although the method is well described, there is a concern that the results in this paper would be difficult for a reader to reproduce.
>
> * **Response**：
>
>   We have submitted a zip file of the source codes in the newly submitted supplementary files. And we promise to open source all the source codes on github to guarantee the reproducibility of the results.
>
> ### - Questions:
>
> > Can you please clarify what is the 'norm' in Equation 6?
>
> * **Response**:
>
>   The norm function in eq. 12 (page 6) refers to the normalization function, which is formulated as follows:
>   $$
>   \begin{equation}
>    X_{\text {norm }}=\frac{X-X_{\min }}{X_{\max }-X_{\min }}
>   \end{equation}
>   $$
>   where $X_{\text {norm }}$ is the normalized data, $X$ is the original data, and $X_{\max }$ and $X_{\min }$ are the maximum and minimum values of the original data, respectively.
>
>   And we apologize for not describing this detail clearly. The soft pseudo labels of the student network are obtained by normalizing the attention scores (to values between 0 and 1) of the teacher network.
>
>   We will revise the statement of ‘attention scores’ on line 11, 64, 156, 186, 320 of the original manuscript to ‘normalized attention scores’, and add the formulation of the normalization function in Section 3.1.2 and Section 3.2.3 in the final version.

---

> ### Author Response · Authors · 2022-08-02
> **Response to Reviewer oxi9 (Part 5/6)**
>
> > 5. The main issue in this work is that it is unclear why there is an improvement? The authors are not using additional supervision. In fact, the pseudo labels are expected to be noisier and change constantly. It is unclear why training a student in parallel helps instance classification even though there are no instance labels. Authors are encouraged to provide further analysis to explain the observed improvement. If I had to guess, the main element that improves performance is in eq.12, in particular the zero-label of instances in negative bags. Authors explicitly provided instance labels but only for negative instances, giving their model a large advantage. Knowing what is not positive helps the model.
>
> * **Response**：
>
>   * **3) From the perspective of information theory, our method avoids a large amount of information loss.**
>
>     Intuitively, if grouping instances into bags is regarded as an operation on the instances, the instance labels have the maximum information entropy, and the bag operation reduces the information in the training data. The information entropy of a bag with instance labels is:
>     $$\begin{equation}
>       H_{I}\left(X_{i}\right)=-K(P \log P+(1-P) \log (1-P))
>     \end{equation} $$
>     The information entropy of a bag with only the bag label is:
>     $$\begin{equation}
>       H_{B}\left(X_{i}\right)=-\left(P^{K} \log P^{K}+\left(1-P^{K}\right) \log \left(1-P^{K}\right)\right)
>     \end{equation} $$
>     , where $K$ represents the bag length and $P$ represents the probability that each instance label is positive. We calculate and visualize the entropy difference between the instance labels and the bag label in Page 7 Section 5 (Rebuttal) Figure 8 of the newly submitted ‘Supplementary Material.pdf’ file. We can see the information loss increases when the bag-length K increases. As a bag usually contains thousands of instances, the bag-based approach will lose a lot of information and is not the best choice for instance-level classification. Even though instance labels from positive bags are not available, our WENO can directly train with all true negative instance labels from negative bags and the soft pseudo labels from the teacher network, thus avoiding information loss in negative bags.
>
>   * **4) For pseudo labels which are “more noisier and constantly changing”:**
>
>     In our method, although the pseudo labels generated by the teacher network are relatively noisy at the beginning, in fact, the pseudo labels become more and more accurate as the training proceeds. Please refer to **Response** of **Weakness 1** for more details about label noise.
>
>   * **5) For the role of zero-labels of instances:**
>
>     True negative instance labels do play an important role, but this is **only one aspect of our framework to achieve substantial improvement**. The strength of our framework comes from combining the bag classification framework and the instance classification framework in a knowledge distillation manner, i. e., the guidance of attention-based soft pseudo labels, the utilization of the information of all true negative instances, the information transfer and feedback of the student and teacher networks, and the use of the hard positive instance mining strategy. The results of the ablation experiments in the following table validate the effectiveness of each component.
>
>     In addition, **using only true negative instances cannot perform the training**.
>
>     | Distillation | Soft Label | Shared Encoder | HPM  | Instance-level AUC | Bag-level AUC |
>     | :----------: | :--------: | :------------: | :--: | :----------------: | :-----------: |
>     |              |            |                |      |       0.8480       |    0.8379     |
>     |      √       |            |                |      |       0.8715       |    0.8516     |
>     |      √       |     √      |                |      |       0.8787       |    0.8574     |
>     |      √       |     √      |       √        |      |       0.9011       |    0.8583     |
>     |      √       |     √      |       √        |  √   |     **0.9271**     |  **0.8663**   |

---

> ### Author Response · Authors · 2022-08-02
> **Response to Reviewer oxi9 (Part 4/6)**
>
> > 4. Hard positive instance mining (hpm) proposed in this paper is similar to hard mining in class-activation maps (CAMs). In the latter, strong activations are suppressed, thereby pushing the network to seek more positive regions. The same here, the authors drop high positive instances simulating suppression. Therefore, the proposed mining strategy is not really new; and literature discussion is necessary. See an example for CAMs ib Wei, Y., Feng, J., Liang, X., Cheng, M.-M., Zhao, Y., and Yan, S. (2017). Object region mining with adversarial erasing: A simple classification to semantic segmentation approach. CVPR 2017.
>
> * **Response**：
>
>   We will add the mentioned literature discussion and discuss the differences in the final version.
>
>   Similar to the mentioned reference, we also use the adversarial erasing strategy (which is very common, e.g., adversarial learning), but **the method of obtaining instance predictions is completely different from that of CAM**. CAM utilizes the network's own attention scores (which in our framework corresponds to the output of the Attention Module in the teacher network), while we use the predictions of the student (instance-level) classifier to perform hard positive instance mining instead of the teacher's own attention scores.
>
>   On the other hand, we introduce the idea of hard positive instance mining into the field of whole slide image processing for the first time, and the results of ablation experiments in Page 9, Section 5, Table 4 of the original manuscript validate the efficiency of this strategy for whole slide image classification.
>
>   | Distillation | Shared Encoder | HPM  | Instance-level AUC | Bag-level AUC |
>   | :----------: | :------------: | :--: | :----------------: | :-----------: |
>   |              |                |      |       0.8480       |    0.8379     |
>   |      √       |                |      |       0.8787       |    0.8574     |
>   |      √       |       √        |      |       0.9011       |    0.8583     |
>   |      √       |       √        |  √   |     **0.9271**     |  **0.8663**   |
>
> > 5. The main issue in this work is that it is unclear why there is an improvement? The authors are not using additional supervision. In fact, the pseudo labels are expected to be noisier and change constantly. It is unclear why training a student in parallel helps instance classification even though there are no instance labels. Authors are encouraged to provide further analysis to explain the observed improvement. If I had to guess, the main element that improves performance is in eq.12, in particular the zero-label of instances in negative bags. Authors explicitly provided instance labels but only for negative instances, giving their model a large advantage. Knowing what is not positive helps the model.
>
> * **Response**：
>
>   * **1) For the performance improvement without using additional supervision:**
>
>     The Multiple Instance Learning-based Whole Slide Image Classification itself is a weakly supervised learning problem with weak labels, and most existing studies investigated how to improve performance without using additional supervision by making better use of available information. Our method utilizes the same supervision as these studies, and we think that the performance improvement comes from the fact that our framework can exploit and mine the available information more efficiently. The most important reason is that we integrate a bag classifier and an instance classifier in a knowledge distillation framework to mutually improve the performance of both classifiers. In addition, we also share the feature extractors and propose the HPM strategy to enhance knowledge distillation to better exploit and mine the available information. As contrast, most existing bag-based methods are only able to utilize the supervised information at the bag-level, while we are able to utilize the normalized attention scores of the teacher network as the soft positive labels for the student network as well as all the negative instance labels.
>
>   * **2) Response to “**Authors explicitly provided instance labels but only for negative instances, giving their model a large advantage**”.**
>
>     In weakly supervised WSI classification, "instance labels for negative instances" is known information, which is available for all methods, but is utilized in different ways by different methods. Thus, the comparison with similar methods is fair. We only use this information more directly.

---

> ### Author Response · Authors · 2022-08-02
> **Response to Reviewer oxi9 (Part 3/6)**
>
> > 2. On page 2, lines 70-72, the authors mentioned that sharing the feature extractor between student and teacher enhances knowledge distillation. It is not clear why it is the case.
>
> * **Response**：
>
>   Sharing the feature extractor plays an important role in the knowledge distillation, so we provide detailed explanation and ablation study in the original manuscript. Concretely, we provide detailed explanations in Page 5, Section 3.2.1 (Framework Overview), lines 158-161 and Page 6, Section 3.2.3 (Student Network), lines 194-199 of the original manuscript. We also present the results of ablation experiments on the Camelyon16 Dataset in Page 9, Section 5 (Ablation Study), Table 4 of the original manuscript to show that sharing feature extractors performs better than using extractors independently.
>
>   We would like to further explain why sharing the feature extractor enhances the knowledge distillation. As shown in Page 3, Figure 1 (a), (b) of the original manuscript, the traditional knowledge distillation methods usually have only single-way knowledge transfer, while the difference between this paper and previous methods is that there is double-way knowledge transfer between the teacher and the student, where the shared feature extractors play an important role. Our teacher and student branches are trained alternately. On this basis, since the feature extractors are shared, the optimization of the extractor when training the teacher can work directly on the student classifier, and the optimization of the extractor when training the student also works directly on the teacher classifier. Since the student classifier learns not only from the teacher's normalized attention scores, but also from the true negative instances, the student's instance-level classification ability can outperform the teacher's attention scores. Through shared feature extractors, the student network transfers this knowledge to the teacher network. On the other hand, the knowledge learned by the teacher network through bag classification and the HPM strategy can also be transferred to the student network, further enhancing the knowledge transfer and feedback.
>
>   In the experimental results, as shown in the second and third rows of Page 9, Section 5, Table 4 of the original manuscript, without HPM, the performance of our framework with sharing extractors alone outperforms the baseline, which confirms the above argument.
>
>   | Distillation | Shared Encoder | HPM  | Instance-level AUC | Bag-level AUC |
>   | :----------: | :------------: | :--: | :----------------: | :-----------: |
>   |              |                |      |       0.8480       |    0.8379     |
>   |      √       |                |      |       0.8787       |    0.8574     |
>   |      √       |       √        |      |       0.9011       |    0.8583     |
>   |      √       |       √        |  √   |     **0.9271**     |  **0.8663**   |
>
>
> > 3. In the entire paper, the pseudo-labels were described as the attention scores. It is not until eq.12 (page 6) that the authors describe the soft labels as being single scores that simulate probability. The text should be clear about this detail from the outset. In addition to fig.2, there should be an operation after attention scores come out from the teacher, before being used by the student’s CE loss.
>
> * **Response**：
>
>   Sorry for not describing this detail clearly. The soft pseudo labels of the student network are obtained by normalizing the attention scores (to values between 0 and 1) of the teacher network. The norm function in eq. 12 (page 6) refers to the normalization function, which is formulated as follows:
>
>   \begin{equation}
>     X_{\text {norm }}=\frac{X-X_{\min }}{X_{\max }-X_{\min }}
>   \end{equation}
>
>   where $X_{\text {norm }}$ is the normalized data, $X$ is the original data, and $X_{\max }$ and $X_{\min }$ are the maximum and minimum values of the original data, respectively.
>
>   We will revise the statement of ‘attention scores’ on line 11, 64, 156, 186, 320 of the original manuscript to ‘normalized attention scores’, and add the formulation of the normalization function in Section 3.1.2 and Section 3.2.3 in the final version.

---

> ### Author Response · Authors · 2022-08-02
> **Response to Reviewer oxi9 (Part 2/6)**
>
> ### - Weaknesses
>
> > 1. On page 2, lines 63-66, the authors mentioned that using soft labels as they did helps to alleviate the noisy pseudo-labels used in other methods. It is not clear how it is the case. From the description of the method, the pseudo-labels in this work change every SGD step since the network that builds them changes constantly. Therefore, they are noisier, especially at the beginning of the training. It is not clear why the authors claimed that their pseudo-labels alleviate the noise issue. In addition, there is no empirical evidence to support this claim.
>
> * **Response**：
>
>   * **Second, we would like to further explain how we alleviate the above problems by using our *soft pseudo-labels*.**
>
>      3) Our student network is also able to learn from all true negative instances, and the student can also transfer the learned knowledge to the teacher through the shared feature extractors. In addition, the proposed Hard Positive Instance Mining (HPM) strategy can also force the teacher to find hard positive instances and give them more accurate pseudo labels. Thus, the pseudo labels generated by the teacher are becoming more accurate rather than noisier.
>
>    * **Third, we would like to explain the** **empirical evidence provided in the original submission and new empirical evidence added in this rebuttal.**
>
> 	  We demonstrate the quality of the pseudo labels in Page 9, Section 5 (Ablation Study), Figure 3 of the original manuscript. Panel (b) in Figure 3 represents the results of the instance-level classification directly using the normalized attention scores of the teacher network. **Note that this result is a direct indicator of the quality of the pseudo labels generated by the teacher.** As shown in the blue line (ABMIL+WENO without HPM) and the red line (ABMIL+WENO with HPM), the pseudo labels generated by ABMIL+WENO are indeed relatively noisy at the beginning of the training, but **the quality of the generated pseudo labels become more and more accurate as the training proceeds.**
>
> 	  There are two main reasons for the increasing quality of WENO's pseudo labels: (1) the teacher (bag-level classifier) itself becomes more well-trained, and its attention scores can gradually distinguish positive/negative instances more accurately, i.e., the pseudo labels will become more accurate. As shown in the gray line in Figure 3 Panel (b), even raw ABMIL eventually achieves an instance-level classification AUC of 0.89. (2) We further adopt the distillation framework of teacher/student networks and adopt the method of sharing parameters so that the knowledge learned by the student can be transferred to the teacher, and we also propose the HPM strategy to force the teacher to learn the hard positive instances. Therefore, as shown in Figure 3 Panel (b), the quality of the final pseudo labels of ABMIL+WENO is much higher than that of raw ABMIL. The above Table 1 validates the role of each proposed modules.
>
> 	  Finally, the soft pseudo labels given by the teacher network do change constantly, but **it does not mean that they become more inaccurate, and since we use soft labels, there is no 0-1 flipping like the hard labels.** As mentioned above, the soft pseudo labels are generally becoming more accurate as the iterations proceed. As shown in Page 9, Section 5 (Ablation Study), Figure 3 of the original manuscript, **the soft pseudo labels given by the teacher network tend to stabilize after the convergence of the teacher and the student network**. Taking the blue line (ABMIL+WENO without HPM) in Figure 3 panel (b) as an example, we **add the variance of the test instance-level AUC of the pseudo labels at each episode** (step size set to 50), as shown in the Table 2 below. It can be seen that the variance of the pseudo labels gradually becomes smaller as the training proceeds.
>
> 	  Table 2: Variance of the Amplitude Change of the Instance Pseudo Labels at Each Episode (step size set to 50).
>
> 	 |      Epoch      |   1-50    |  51-100   |  101-150  |  151-200  |  201-250  |  251-300  |
> 	 | :-------------: | :-------: | :-------: | :-------: | :-------: | :-------: | :-------: |
> 	 | ABMIL[1] + WENO | 3.804E-02 | 5.995E-06 | 9.418E-07 | 7.203E-07 | 2.247E-07 | 1.647E-07 |
>
>
> [1] Ilse et al. "Attention-based deep multiple instance learning." ICML, 2018.
>
> [2] Li et al. "Dual-stream multiple instance learning network for whole slide image classification with self-supervised contrastive learning." CVPR. 2021.
>
> [3] Zhang et al. "DTFD-MIL: Double-Tier Feature Distillation Multiple Instance Learning for Histopathology Whole Slide Image Classification." CVPR. 2022.
>
> [4] Hinton et al. "Distilling the knowledge in a neural network." arXiv preprint arXiv:1503.02531 2.7 (2015).

---

> ### Author Response · Authors · 2022-08-02
> **Response to Reviewer oxi9 (Part 1/6)**
>
> Thank you very much for your valuable comments, which are very helpful to clarify details and improve the quality of our paper. We will respond to all the weaknesses and questions in detail. Some figures and results of newly added experiments can be found in Page 6, Section 5 (Rebuttal) in the newly submitted ‘Supplementary Material.pdf' file.
>
> ### - Weaknesses
>
> > 1. On page 2, lines 63-66, the authors mentioned that using soft labels as they did helps to alleviate the noisy pseudo-labels used in other methods. It is not clear how it is the case. From the description of the method, the pseudo-labels in this work change every SGD step since the network that builds them changes constantly. Therefore, they are noisier, especially at the beginning of the training. It is not clear why the authors claimed that their pseudo-labels alleviate the noise issue. In addition, there is no empirical evidence to support this claim.
>
> * **Response**：
>
>   * **First, we would like to further explain what does “noisy pseudo labels” mean on Page 2, lines 63-66.** In weakly-supervised WSI analysis tasks, we only have the labels of slides (bags) and do not have the labels of patches (instances). In this setting, when we tackle the problem with instance-based methods, which first train an instance classifier and then aggregate the predictions of each instance in a bag to obtain the bag prediction, we need pseudo labels of the instances for model training. However, since the true labels of instances in positive bags are unknown, these methods typically train the instance classifier by selecting instances from positive bags and directly assigning them the hard positive pseudo labels (e.g., value 1). **This way of assigning pseudo labels leads to a large number of errors (which we denote as label noise) in the assigned positive pseudo labels**, thus limiting their performance. By "alleviate the noisy pseudo-labels", we mean that **our method can provide more accurate pseudo labels for training the student network**. There are two reasons contributing to the noisy-pseudo-label problem in instance-based methods: one is that the pseudo labels of these methods are usually produced by the instance classifier, which is also trained only with these pseudo labels. When the instance pseudo labels are wrong, the instance classifier will also give wrong predictions and thus falls into an error loop. Second, these methods adopt the hard-pseudo-label assignment strategy, i.e., directly assigning hard positive labels (e.g., value 1) to the selected positive instances. This also worsens the problem of incorrect pseudo labels.
>
>   * **Second, we would like to further explain how we alleviate the above problems by using our *soft pseudo-labels*.**
>
> 	  1) The positive pseudo labels of instance classifier are distilled from the normalized attention scores of the teacher network. The teacher network is trained with correct bag labels, and the scores of each instance from the positive bags are delivered to the student network by normalized attention scores. In existing bag-based methods using attention mechanism, attention scores are widely used to produce instance predictions [1-3]. Thus, the student network gets part of its supervision from the teacher network, which helps reduce noise. As the training proceeds, **the pseudo labels generated by the teacher are actually more and more accurate rather than noisier.**
>
> 	  2) Normalized attention scores are a kind of soft labels. As mentioned in [4], soft labels contain more useful information compared to hard labels and are a very effective way of transferring knowledge between models. Using the normalized attention scores of the teacher network as the soft pseudo labels for the student network can avoid the effect of wrong hard pseudo labels. **We further added a comparison experiment on soft labels vs. hard labels** in the ablation study on Page 9, Section 5 (Ablation Study), Table 4 of the original manuscript, and the results are shown in the following Table 1. These results show that **using soft pseudo labels works better than using hard pseudo labels in our distillation framework**.
>
> 	     Table 1: Ablation study on the Camelyon16 Dataset, where the second row indicates distillation using hard pseudo labels (true negative labels as well as pseudo labels that assign all instances in the positive bags to 1), and the third row indicates distillation using soft pseudo labels (true negative labels as well as normalized attention scores from the teacher network as positive pseudo labels).
>
>          | Distillation | Soft Label | Shared Encoder | HPM  | Instance-level AUC | Bag-level AUC |
>          | :-: | :-: | :-: | :--: | :-: | :-: |
>          |||||0.8480|0.8379|
>          |√||||0.8715       |    0.8516     |
>          |√|√|||       0.8787       |    0.8574     |
>          |√|√|√        |      |       0.9011       |    0.8583     |
>          |√|√|       √        |  √   |     **0.9271**     |  **0.8663**   |

---

> ### Comment · Reviewer_oxi9 · 2022-08-08
> **Response to Rebuttal**
>
> I thank the authors for their detailed response.  My rating for that paper has been increased to a score of 6  (weak accept).

---

### Official Review · Reviewer_S7zj · 2022-07-11

**Rating:** 8
**Confidence:** 4
**Soundness:** 4 excellent
**Presentation:** 4 excellent
**Contribution:** 4 excellent

**Summary:**

The authors utilization whole slide image (WSI) classification as a classic instance and bag classification problem as most tasks based on whole slide images have slide-level (bag) classification labels and small patches which compose the slides (instances) for the the instance labels are unknown.  This formulation is necessary because WSI contain too much data to be process on a single GPU. Multiple instances learning paired with an aggregator function, like RNN, has been proven to deliver clinical level results, most classically for "tumor/not tumor" tasks.  The authors suggest that these method are limited in tasks where instance level classifications are dichotomized as challenging and not challenging and that certain bags with few levels of challenging instances can be false negatives.  All of this is a set up for their model which benefits from work in the field of knowledge distillation.   Knowledge distillation is a pairing of two models labeled student and teacher, where the teacher network is a MIL-type model using bags of instances with attention and the the student is an instance level classifier that uses all instances in negative bags as negatives and instances with "positive attention" from the teacher network labeled as positives (authors call them soft pseudo labels).  The authors also employ a "Hard Positive Instance Mining" HPM strategy whereby the instances where are predicted to be positive with high probability are excluded from the teacher training to make teacher task more challenging. The authors then implemented the strategy using synthetic datasets with CIFAR-10 and CRC-100K datasets with instance labels.  Camelyon16 was used as a real cancer dataset with both instance and bag level label. Finally, the model was trained and tested on real-world datasets without known instance level classifications, TCGA NSCLC dataset and and in house cervical cancer dataset.  Using both ABMIL and DSMIL, adding knowledge distillation improved overall bag level classification performance as measured by AUC.

**Questions:**

This method appears to heighten the negative consequences of incorrectly labeled data (specifically true positives false labeled negative) because those instances labeled negative are going to adversely train the student network. Have you considered pairing this work with networks trained to catch this situation?

**Limitations:**

The major weakness of the paper is that the methods has not been used on well-benchmarked classification tasks.  For example, does this strategy actually enhance methodologies for which MIL+RNN already deliver clinical grade results or is this method only going to improve results for which the dataset is small and/or the task is not clinically significant.

**Strengths And Weaknesses:**

This work is very strong because the authors have properly formulated the many challenges of whole slide image classification and incorporated emerging methods from the field of knowledge distillation/transfer learning to improve upon existing state of the art methodologies in WSI classifications.  From my own work, I have observed that attention maps derived from MIL training provide what appears to be very useful information but that information, as far as I know, has not been effectively harnessed in a practical way.  By formulating the problem with interactions between student and teacher networks, the bag level classifier can provide soft labels for the classification of tiles while the classification results from the student network can be harnessed to "stress test" the bag level classifier.

While the authors describe in detail the concept of HPM, the study and results are very weakly developed and presented.  Thus I think that perhaps including this as part of the work is a bit premature and the remaining work to my eyes, stands on its own two feet.

---

> ### Author Response · Authors · 2022-08-02
> **Response to Reviewer S7zj (Part 3/3)**
>
>
>   ### - Limitations
>
> > The major weakness of the paper is that the methods has not been used on well-benchmarked classification tasks. For example, does this strategy actually enhance methodologies for which MIL+RNN already deliver clinical grade results or is this method only going to improve results for which the dataset is small and/or the task is not clinically significant.
>
> * **Response**：
>
>   Thank you for your comments. Actually, besides the two synthetic datasets, WENO was carefully validated on **three real-world clinical datasets**, including two widely used public benchmark datasets in the WSI field [2,3,4], the Camelyon16 Dataset and the TCGA Lung Cancer Dataset.
>
>   **About clinically significant tasks**. To further demonstrate the potential of WENO to be applied to clinically significant problems, we experimented on **an inhouse cervical cancer dataset**. This task is to predict the lymph node metastatic status (positive or negative) of a patient from HE slides of the primary lesion. **Clinically**, this is a rarely studied but clinically significant task in WSI analysis field. Cervical cancer is the most common malignant tumor of the female reproductive system [5]. Accurate diagnosis of lymph node metastatic status is crucial to the selection of individualized treatment for patients with cervical cancer [6,7,8]. So far, there is still a lack of effective method to assess the lymph node metastatic status before surgical treatment. On the contrary, the HE slides of the primary lesion can be obtained by biopsy before surgical treatment. **Technically**, unlike the two former real-world datasets, this task is particularly challenging. Since there is no prior knowledge about what image features of the primary lesion indicate metastasis, even very experienced pathologists are unable to accurately distinguish between positive and negative instances of the HE slides of primary lesion. In this task, WENO achieves the best performance among all comparison methods, which indicates that WENO can be applied to the prediction of metastasis using primary lesion WSIs in clinical practice.
>
>   **Performance comparison to MIL+RNN.** Comparison to MIL+RNN was only done on the TCGA dataset in the original manuscript and we **added comparison on the other two real-world datasets**. As can be seen, using WENO outperformed **MIL+RNN**[9] on all three clinical datasets. The results are presented in the following tables.
>
>   - **Results on the Camelyon16 Dataset:**
>
>      |Methods|Instance-level AUC|Bag-level AUC|
>      |------------------|:------------------:|:-------------:|
>      |MIL-RNN [9]|0.8245|0.8162|
>      |ABMIL [1] + WENO|0.9271|**0.8663**|
>      |DSMIL [2] + WENO|**0.9377**|0.8495|
>
>   - **Results on the TCGA Lung Cancer Dataset:**
>
>      |Methods|Bag-level AUC|
>      |-----------------|:-------------:|
>      |MIL-RNN [9]|0.9107|
>      |ABMIL[1] + WENO|0.9663|
>      |DSMIL[2] + WENO|**0.9727**|
>
>   - **Results on the Cervical Cancer Dataset:**
>
>      |Methods|Bag-level AUC|
>      |-----------------|:-------------:|
>      |MIL-RNN [9]|0.7563|
>      |ABMIL[1] + WENO|0.8056|
>      |DSMIL[2] + WENO|**0.8222**|
>
>   **Potential of being used on large datasets.** Finally, WENO was not validated on the large clinical datasets mentioned in the MIL+RNN paper [9], because most of the data in that study are not publicly available [9] (mentioned in Section Data availability). However, given the excellent performance demonstrated on the three clinical datasets and the fact that WENO does not increase the complexity of existing algorithms much, WENO could be used on large clinical datasets.
>
>
> [1] Ilse et al. "Attention-based deep multiple instance learning." ICML. 2018.
>
> [2] Li et al. "Dual-stream multiple instance learning network for whole slide image classification with self-supervised contrastive learning." CVPR. 2021.
>
> [3] Shao et al. "Transmil: Transformer based correlated multiple instance learning for whole slide image classification."  NeurIPS 34 (2021): 2136-2147.
>
> [4] Zhang et al. "DTFD-MIL: Double-Tier Feature Distillation Multiple Instance Learning for Histopathology Whole Slide Image Classification." CVPR. 2022.
>
> [5] COHEN et al. Cervical cancer [J]. Lancet, 2019,393(10167):169-182.
>
> [6] CANFELL et al. Mortality impact of achieving WHO cervical cancer elimination targets: a comparative modelling analysis in 78 low-income and lower-middle-income countries [J]. Lancet, 2020,395(10224):591-603.
>
> [7] ZIGRAS et al. Early Cervical Cancer: Current Dilemmas of Staging and Surgery [J]. Curr Oncol Rep, 2017,19(8):51.
>
> [8] LEE et al. 2018 FIGO Staging System for Uterine Cervical Cancer: Enter Cross-sectional Imaging [J]. Radiology, 2019,292(1):15-24.
>
> [9] Campanella et al. "Clinical-grade computational pathology using weakly supervised deep learning on whole slide images." Nature Medicine 25.8 (2019): 1301-1309.

---

> ### Author Response · Authors · 2022-08-02
> **Response to Reviewer S7zj (Part 2/3)**
>
> ### - Questions
>
> > This method appears to heighten the negative consequences of incorrectly labeled data (specifically true positives false labeled negative) because those instances labeled negative are going to adversely train the student network. Have you considered pairing this work with networks trained to catch this situation?
>
> * **Response**：
>
>   Indeed, there are still some incorrectly labeled instances in the pseudo labels used for training the student network (an instance classifier) and we agree that pairing this work with networks trained to catch this situation is a very promising research direction. We will consider this combination in our future work to reduce the influence of the noisy pseudo labels produced by the teacher and further improve the performance of both networks. Thank you for your suggestion.
>
>   Actually, in weakly supervised WSI classification, we always face the problem of noisy label in training an instance classifier because only the bag-level labels are available while the instance-level labels are not. In this paper, we tackle this problem by **reducing the noise of the pseudo labels by constructing a distillation framework**. We used a simple instance classifier as the student network, while other more advanced instance classifiers can also be used as the instance classifier in our framework. While we do not think that our method **heightens the negative consequences of incorrectly labeled data**, we agree that if an instance classifier that is more robust to incorrectly labeled instances in training data is used as the student network, the overall performance can potentially be further improved.

---

> ### Author Response · Authors · 2022-08-02
> **Response to Reviewer S7zj (Part 1/3)**
>
> Thank you very much for the valuable comments, which are very helpful to further improve our manuscript. We will respond to the weaknesses, questions and limitations in detail. Some figures and results of newly added experiments can be found in Page 6, Section 5 (Rebuttal) in the newly submitted ‘Supplementary Material.pdf' file.
>
> ### - Weaknesses
>
> > While the authors describe in detail the concept of HPM, the study and results are very weakly developed and presented. Thus I think that perhaps including this as part of the work is a bit premature and the remaining work to my eyes, stands on its own two feet.
>
> * **Response**：
>
>   We appreciate your comments on the HPM strategy, and we agree that the HPM strategy deserves in-depth study. In the original manuscript, we have done the following three studies on the HPM strategy: **1)** In Page 9, Section 5, we present an ablation study (Table 4) to show the effectiveness of HPM; **2)** In Figure 3, we show how the HPM strategy influences the performance of the teacher and the student networks for instance classification; **3)** In Supplementary Material Page 5, Section 3, we conduct a robustness study on the threshold of the HPM strategy on the Camelyon16 Dataset, and the results are given in Table 3 of the Supplementary Material. To further study the HPM strategy, we perform **a new experiment about the robustness of HPM on its starting epoch**, and the results can be found in Page 6 Section 5 (Rebuttal) Figure 7 in the newly submitted ‘Supplementary Material.pdf’. Generally speaking, the HPM strategy is robust to the threshold and the starting epoch. We copy the main results of these studies here for your convenience.
>
>   * **(1)**  **Ablation Study of HPM Strategy on the Camelyon16 Dataset**
>
>   	As shown in the third and fourth rows of the following Table (Page 9, Section 5, Table 4 in the original manuscript), **adding HPM can improve the AUC of both instance-level and bag-level classification**.
>
>     | Distillation | Shared Encoder | HPM  | Instance-level AUC | Bag-level AUC |
>     | :----------: | :------------: | :--: | :----------------: | :-----------: |
>     |       |    |     |  0.8480 | 0.8379 |
>     |  √   |    |     |  0.8787 | 0.8574 |
>     |  √   | √  |    |  0.9011 | 0.8583 |
>     |  √   | √  | √ |  **0.9271** |  **0.8663** |
>
>   * **(2) About how the HPM strategy influences the performance of the teacher and the student networks**
>
>   	In Page 9, Section 5, Figure 3 of the original manuscript, comparing the blue line (ABMIL+WENO without HPM) and the red line (ABMIL+WENO with HPM) in panel (b) and panel (c), we can see that the instance classification performance of both the teacher network and the student network **steadily improves after some vibration when HPM is added**.
>
>   * **(3)**  **Robustness Study of the Threshold of the HPM Strategy on the Camelyon16 Dataset**
>
>   	As can be seen in the following table (Supplementary Material Page 5, Section 3, Table 3), the instance-level AUC with HPM strategy is significantly higher than that without HPM strategy at all thresholds; when the threshold is below 0.92, the bag-level AUC with HPM strategy is slightly lower than without HPM strategy, which may be caused by dropping too many positive instances. The instance-level AUC and bag-level AUC using HPM strategy are higher than the raw ABMIL at all thresholds. These results show that the **HPM strategy is robust to the Threshold used.**
>
>     | Method| Threshold | Instance-level AUC | Bag-level AUC |
>     | -------------------- | :-------: | :----------------: | :-----------: |
>     | ABMIL [1]| - | 0.8480|0.8379|
>     | ABMIL+WENO (w/o HPM) |1.0000| 0.8787|0.8574|
>     | ABMIL+WENO (w/ HPM)  |0.9800| 0.8884 | **0.8717** |
>     | ABMIL+WENO (w/ HPM)  |0.9600| 0.9196 | 0.8675 |
>     | ABMIL+WENO (w/ HPM)  |0.9400|  **0.9271** | 0.8663 |
>     | ABMIL+WENO (w/ HPM)  |0.9200| 0.9256 | 0.8402 |
>     | ABMIL+WENO (w/ HPM)  |0.9000| 0.9204 | 0.8495 |
>     | ABMIL+WENO (w/ HPM)  |0.8800| 0.9042 | 0.8478 |
>
>   * **(4)**  **Robustness Study of the Starting Epoch of the HPM Strategy on the CIFAR-10-MIL Dataset**
>
>   	We added a new robustness study on the starting epoch of the HPM strategy, and the results are shown in Page 6 Section 5 (Rebuttal) Figure 7 of the newly submitted ‘Supplementary Material.pdf’ file. We construct the WENO framework using ABMIL [1] as the teacher and experimented on a CIFAR-10-MIL dataset with a positive instance ratio of 0.2. In this experiment, we added the HPM strategy at three different time points (the 100th, 150th and 200th epoch) and the results show that their final instance-level AUCs are almost the same for both the teacher and the student networks. These results indicate that **HPM is not sensitive to the starting epoch.**
>
> [1] Ilse, Maximilian, Jakub Tomczak, and Max Welling. "Attention-based deep multiple instance learning." International Conference on Machine Learning. PMLR, 2018.

---

### Meta-Review · Area_Chair_J5pQ · 2022-08-26

**Recommendation:** Accept
**Confidence:** Certain

**Metareview:**

This submission was reviewed by three reviewers. All three reviewers provided detailed comments during the review period. The authors provided detailed responses to the initial set of reviews. The rebuttals lead to improved scores of some reviewers while other reviewers confirmed that their concerns have been addressed. Given the above evaluations and interactions, an accept is recommended.

**Award:**

No

---

### Decision · Program_Chairs · 2022-09-14

Accept